# Nasal airway transcriptome-wide association study of asthma reveals genetically driven mucus pathobiology

Satria P. Sajuthi[1], Jamie L. Everman[1], Nathan D. Jackson[1], Benjamin Saef[1], Cydney L. Rios[1], Camille M. Moore[1,2,3], Angel C. Y. Mak[4], Celeste Eng[4], Ana Fairbanks-Mahnke[1], Sandra Salazar[4], Jennifer Elhawary[4], Scott Huntsman[4], Vivian Medina[5], Deborah A. Nickerson[6], Soren Germer[7], Michael C. Zody[7], Gonçalo Abecasis[8], Hyun Min Kang[8], Kenneth M. Rice[9], Rajesh Kumar[10], Noah A. Zaitlen[11], Sam Oh[4], NHLBI Trans-Omics for Precision Medicine (TOPMed) Consortium*, José Rodríguez-Santana[5], Esteban G. Burchard[4,12] & Max A. Seibold[1,13,14 ✉]

To identify genetic determinants of airway dysfunction, we performed a transcriptome-wide association study for asthma by combining RNA-seq data from the nasal airway epithelium of 681 children, with UK Biobank genetic association data. Our airway analysis identified 102 asthma genes, 58 of which were not identified by transcriptome-wide association analyses using other asthma-relevant tissues. Among these genes were *MUC5AC*, an airway mucin, and *FOXA3*, a transcriptional driver of mucus metaplasia. Muco-ciliary epithelial cultures from genotyped donors revealed that the *MUC5AC* risk variant increases MUC5AC protein secretion and mucus secretory cell frequency. Airway transcriptome-wide association analyses for mucus production and chronic cough also identified *MUC5AC*. These cis-expression variants were associated with trans effects on expression; the *MUC5AC* variant was associated with upregulation of non-inflammatory mucus secretory network genes, while the *FOXA3* variant was associated with upregulation of type-2 inflammation-induced mucus-metaplasia pathway genes. Our results reveal genetic mechanisms of airway mucus pathobiology.

[1] Center for Genes, Environment, and Health, National Jewish Health, Denver, CO, USA. [2] Department of Biomedical Research, National Jewish Health, Denver, CO, USA. [3] Department of Biostatistics and Informatics, University of Colorado, Denver, CO, USA. [4] Department of Medicine, University of California-San Francisco, San Francisco, CA, USA. [5] Centro de Neumología Pediátrica, San Juan, PR, USA. [6] Department of Genome Sciences, University of Washington, Seattle, WA, USA. [7] New York Genome Center, New York, NY, USA. [8] Center for Statistical Genetics, University of Michigan, Ann Arbor, MI, USA. [9] Department of Biostatistics, University of Washington, Seattle, WA, USA. [10] Ann and Robert H. Lurie Children's Hospital of Chicago, Department of Pediatrics, Northwestern University, Chicago, IL, USA. [11] Department of Neurology and Computational Medicine, University of California Los Angeles, Los Angeles, CA, USA. [12] Department of Bioengineering and Therapeutic Sciences, University of California-San Francisco, San Francisco, CA, USA. [13] Department of Pediatrics, National Jewish Health, Denver, CO, USA. [14] Division of Pulmonary Sciences and Critical Care Medicine, University of Colorado School of Medicine, Aurora, CO, USA. *A list of authors and their affiliations appears at the end of the paper. ✉email: seiboldm@njhealth.org

Asthma is a complex phenotype, composed of multiple pathobiological subgroups (i.e., endotypes), which are determined by poorly understood interactions between clusters of genetic and inhaled environmental risk factors[1,2]. At the tissue level, these gene by environment interactions occur in the airway epithelium, which forms the primary interface through which the airways encounter these inhaled factors. For example, severe viral respiratory illnesses in early life are associated with development of childhood asthma among subjects carrying risk variants at the 17q21 locus[3], which contains multiple genes expressed in the airway epithelium (ORMDL3, GSDMB, GSDMA), although it is unclear through which gene(s) asthma risk is mediated[3–5]. Additionally, respiratory viruses and allergens can trigger airway epithelial production of potent cytokines such as IL-33 and TSLP, whose genes harbor strong genetic risk loci for asthma[6–9]. Neither the causative variants for these loci nor their genetic mechanisms of effect are known.

Importantly, IL-33 and TSLP recruit to the airway and activate immune cells (e.g. mast cells, eosinophils), resulting in their production of type 2 (T2) cytokines (IL-4, IL-5, IL-13) that then signal back to the epithelium, resulting in a feedback loop of T2 inflammation. Approximately 50% of asthmatics exhibit persistent bronchial airway T2 inflammation (i.e., T2-high), forming the most prevalent endotype of asthma identified to date[10]. Specifically, IL-13 stimulation of both mouse and human airway epithelia is sufficient to drive increases in the expression of MUC5AC, an airway mucin gene, and mucus-metaplastic transformation of the tissue[11,12], both of which contribute to the airway obstruction seen in asthma. Although variants in the airway gel-forming mucin locus (chr:11p15) were recently associated with adult onset asthma[9], suggesting a role for genetics in mucus dysregulation, the specific gene(s) driving this association and the mechanism of effect have not been determined.

Transcriptome-wide association studies (TWAS) provide a statistical framework to identify cis-regulated gene expression that is associated with traits[13]. This method allows the potential gene, its regulatory mechanism, and the affected tissue driving GWAS loci to be determined. Ideally, TWAS gene expression models should be developed in the tissue(s) of pathogenesis for the disease being studied. As such, TWAS analysis of the bronchial airway epithelium holds great promise for the identification of genes and mechanisms involved in asthma. However, obtaining enough bronchial epithelial samples to adequately power a TWAS has been challenging given that collection of these samples necessitates performance of invasive bronchoscopies. In contrast, the nasal airway epithelium, which is a bronchial surrogate tissue, can be collected in a minimally invasive manner. The nasal airway epithelium contains the same cell types and is subject to the same inhaled exposures as the bronchial airway epithelium. Moreover, we have shown that the nasal airway epithelium expresses the same genes as the bronchial airway epithelium, with highly correlated expression levels between these airway sites[14]. Importantly, we have also shown that signatures of both asthma and T2-high inflammation can be observed using nasal expression data[14,15].

Here, we use whole transcriptome and whole genome sequencing data from the nasal airway epithelium of 681 children who are asthmatic (n = 434) or healthy controls (n = 247), recruited as part of the Genes-Environments and Admixture in Latino Americans (GALA II) cohort, to identify airway epithelial cis-eQTL variants. We then use these data to build genetic models of airway gene expression which form the basis of TWAS analyses of both adult- and childhood-onset asthma (AOA, COA), drawing on GWAS data from the UK Biobank. We reveal plausible airway genes and mechanisms underlying a large number of asthma GWAS loci. Importantly, we discover genetic mechanisms

of asthma risk for the airway mucin gene, MUC5AC, and the T2 mucus metaplasia transcription factor, FOXA3. Moreover, we reveal how these cis-variants mediate trans-effects involving distinct mucus secretory networks and functionally validate their effects in vitro. Finally, we find that the MUC5AC cis-variant mediates increased MUC5AC protein secretion into mucus while also being a risk factor for self-reported mucus production and chronic cough.

## Results

**The nasal airway epithelium expresses a unique transcriptome that exhibits widespread genetic regulation.** We first sought to determine the uniqueness of the nasal airway epithelium transcriptome, generated from 695 children (asthmatic = 441, healthy controls = 254) in the GALA II cohort, with respect to 26 diverse tissues procured and analyzed by the GTEx consortium[16]. We performed multi-dimensional scaling (MDS) of the top 10,000 most variable genes across all tissues. Plotting the top two dimensions from this analysis largely clustered samples by tissue type and revealed that nasal airway epithelial samples were separated from other tissues in MDS space (Fig. 1a). In fact, hierarchical clustering of the expression profiles from these tissues found that the nasal epithelium, along with testis, exhibited the greatest distance from other tissues (Fig. 1b, top). Additionally, of the top 1000 most variable genes for each tissue, the pairwise overlap of these genes between the nasal epithelium and all other tissues tended to be lower (median: 182 genes) than for all other pairwise GTEx tissue combinations (median: 315 genes, Fig. 1b, bottom).

We performed cis-eQTL mapping of the GALA II nasal airway epithelium transcriptome data using DNA variants obtained from whole genome sequencing data from 681 of the children (asthmatic = 434, healthy controls = 247), analyzing only variants with a minor allele frequency (MAF) greater than one percent. This analysis identified 4,032,782 eQTL variants (eVariants), which were reduced to 34,725 independent eVariants through forward-backward stepwise regression analysis (Fig. 1c, Supplementary Data 1). These eVariants were associated with the expression of 13,807 genes (eGenes, Supplementary Data 2), meaning that 81% of genes expressed in the nasal epithelium are under some level of genetic control. This widespread genetic regulation was maintained when restricting analysis to the 14,894 annotated protein coding genes expressed in the nasal epithelium, of which 12,215 (82%) were eGenes. Average heritability of expression among nasal eGenes was 25%, with 11% of eGenes demonstrating a heritability of greater than 50% (Fig. 1d). Of eGenes in the nasal epithelium, we found 72% had more than one independent eQTL variant, with 30%, 21%, and 21% of eGenes exhibiting 2, 3, or ≥4 independent eQTL variants, respectively (Fig. 1e). Finally, we found that the number of eVariants harbored by an eGene was positively correlated with its heritability (r = 0.6, p < 2.2e-16, Fig. 1f). Together, these results reveal that the nasal epithelium exhibits a unique transcriptome that is strongly regulated by a large set of common genetic variants.

**Nasal epithelium TWAS identifies COA/AOA genes.** Combining our eQTL data from the nasal epithelium with published summary statistics from a recent GWAS[6] analysis for both childhood and adult onset asthma (COA and AOA, respectively) in the UK Biobank population (13,962 COA subjects, 26,582 AOA subjects, 300,671 controls), we performed a TWAS analysis using the FUSION[13] software package. We identified 93 significant TWAS genes (Bonferroni-corrected p value threshold = 3.99e–6) for COA and 21 genes for AOA, for a total

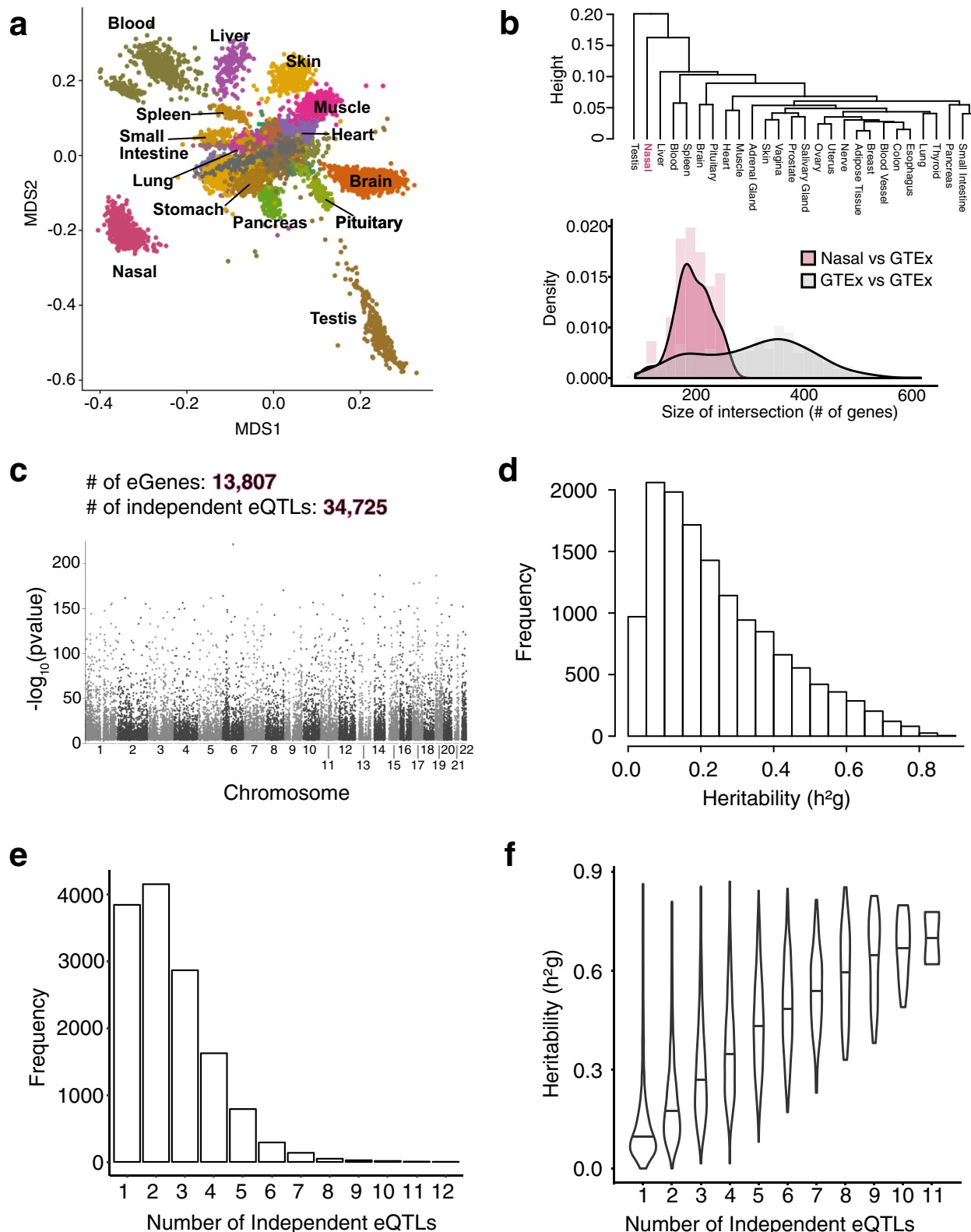

**Fig. 1 The nasal airway epithelium exhibits a unique expression profile that is under strong genetic regulation. a** Samples from 26 GTEx consortium tissues/organs and GALA II nasal airway epithelium datasets ($n = 695$) are plotted based on values for the first two dimensions of an MDS analysis performed on the top 10,000 most variable genes in the combined datasets. **b** Top: Tissue dendrogram based on hierarchical clustering of the tissue median expression values for all genes. Bottom: Density plot of the intersection size between the top 1000 most variable genes in the nasal tissue and other GTEx tissues (pink) and intersection size among GTEx tissues (gray). **c** The nasal airway epithelium eQTL Manhattan plot of tested genome-wide genetic variants ($n = 681$). P values were obtained from two-sided tests. **d** Histogram of nasal eGene heritability ($h^2_g$) estimates. **e** Histogram of the number of independent eQTLs per nasal eGene. **f** Violin plot of nasal eGene heritability stratified by the number of independent eQTLs.

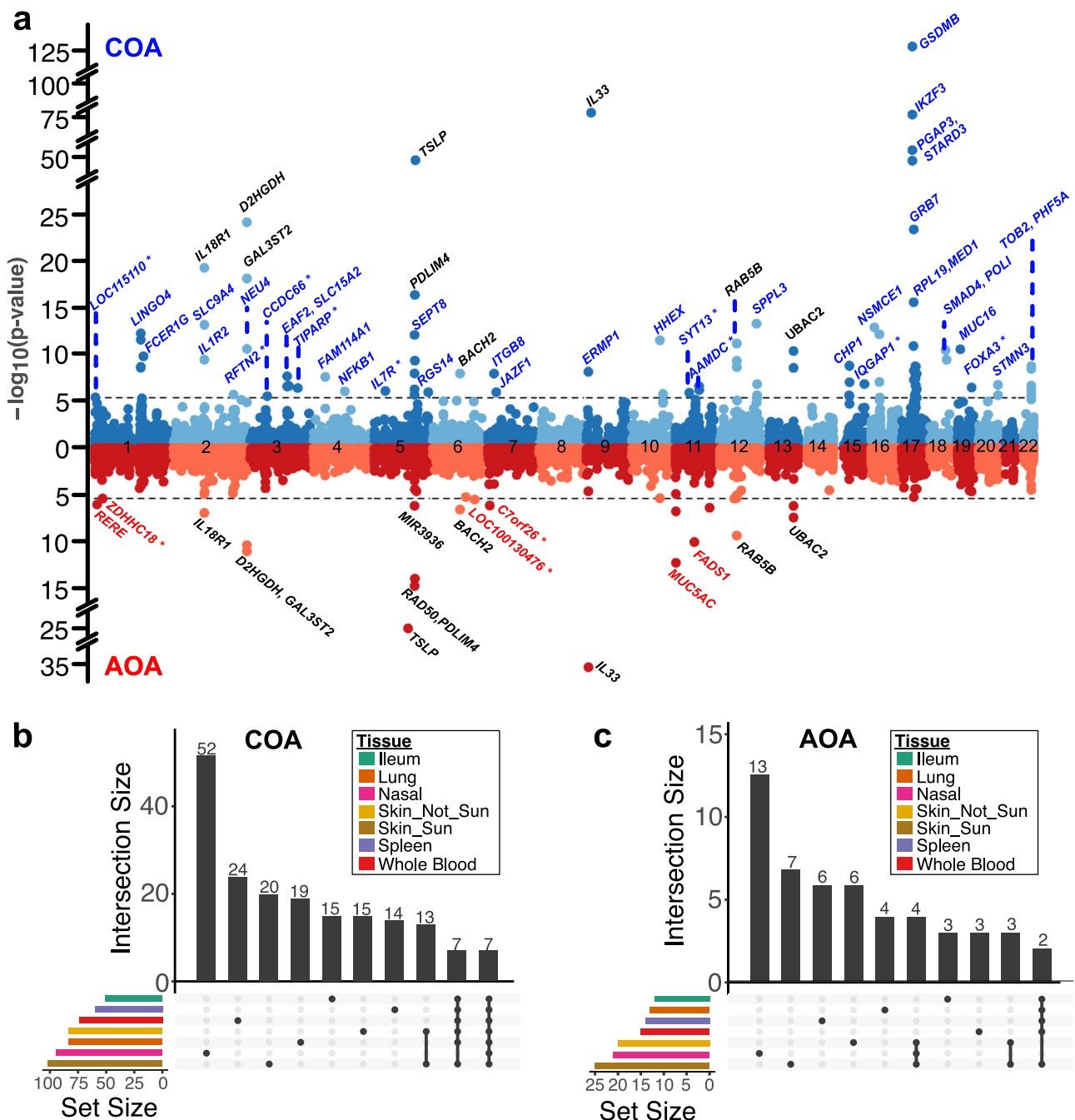

**Fig. 2 A nasal airway epithelium TWAS identifies tissue-specific COA/AOA genes. a** Miami plot of COA (blue) and AOA (red) TWAS results using nasal airway epithelium gene expression models. Select significant genes are highlighted (blue: COA specific, red: AOA specific, black: shared). Genes (not identified by GWAS) are indicated by an asterisk (*). TWAS *p* values were obtained from two-sided tests. **b** Upset plots displaying the number of unique and overlapping significant genes between different tissue COA TWAS analyses. **c** Upset plots displaying the number of unique and overlapping significant genes between different tissue AOA TWAS analyses.

of 102 unique genes (12 shared between COA and AOA, Fig. 2a, Table 1). The higher number of TWAS genes for COA reflected the higher number of independent risk variants identified by the GWAS. We identified at least one nasal TWAS gene in close proximity (1 Mb) to 33 of the 89 independent COA risk loci (37%) and to 13 of the 40 independent AOA risk loci (33%, Supplementary Data 3). We also identified 9 COA TWAS genes (*SYT13, IQGAP1, FOXA3, TIPARP, LOC115110, CCDC66, AAMDC, RFTN2,* IL7R) and 3 AOA TWAS genes (*ZDHHC18, C7ORF26, LOC100130476*) that were not within a Mb of any GWAS risk variant (Fig. 2a).

To evaluate the extent that asthma TWAS genes in the nasal epithelium are unique to that tissue, we also compared our nasal TWAS results to those we generated using other asthma-relevant GTEx tissues. Specifically, we performed TWAS analysis with 6 other tissues: skin (sun-exposed and non-exposed), which is another epithelial tissue; ileum, which contains a mucosal epithelium; blood and spleen, two immune cell-containing tissues; and whole lung, very similar to the another COA/AOA GTEx/UK Biobank TWAS performed recently[9]. TWAS analysis of these tissues generated between 12 and 25 AOA risk genes and between 50 and 100 COA risk genes (Supplementary Data 4).

**Table 1 AOA and COA nasal epithelium TWAS genes.**

| No. | Gene | Position (Hg38) | HSQ | Childhood Onset Asthma | | | | Adult Onset Asthma | | | | Nasal Specific? |
|---|---|---|---|---|---|---|---|---|---|---|---|---|
| | | | | TWAS.Z | TWAS.P | PP3 | PP4 | TWAS.Z | TWAS.P | PP3 | PP4 | |
| **Childhood Onset Asthma** | | | | | | | | | | | | |
| 1 | LOC115110 | 1:2549920-2557011 | 0.46 | 4.63 | 3.73E-06 | 0.02 | 0.98 | 3.15 | 1.65E-03 | NA | NA | Y |
| 2 | RIIAD1 | 1:151721537-151729606 | 0.54 | -5.99 | 2.09E-09 | 1.00 | 0.00 | -1.87 | 6.12E-02 | NA | NA | Y |
| 3 | MRPL9 | 1:151759643-151763916 | 0.14 | -5.96 | 2.45E-09 | 1.00 | 0.00 | -1.70 | 8.82E-02 | NA | NA | Y |
| 4 | LINGO4 | 1:151800265-151805442 | 0.42 | 7.21 | 5.39E-13 | 1.00 | 0.00 | 0.88 | 3.77E-01 | NA | NA | N |
| 5 | FLG-AS1 | 1:152213459-152366692 | 0.31 | -7.00 | 2.60E-12 | 1.00 | 0.00 | -1.52 | 1.29E-01 | NA | NA | N |
| 6 | FCER1G | 1:161215297-161219248 | 0.19 | 6.40 | 1.57E-10 | 0.00 | 1.00 | 3.70 | 2.12E-04 | NA | NA | N |
| 7 | IL1R2 | 2:101991844-102028422 | 0.07 | -6.27 | 3.57E-10 | 0.99 | 0.00 | -3.81 | 1.41E-04 | NA | NA | Y |
| 8 | SLC9A4 | 2:102473303-102533972 | 0.21 | -7.50 | 6.55E-14 | 1.00 | 0.00 | -4.38 | 1.18E-05 | NA | NA | Y |
| 9 | RFTN2 | 2:197570803-197675860 | 0.31 | -4.77 | 1.86E-06 | 0.07 | 0.93 | -4.06 | 4.87E-05 | NA | NA | N |
| 10 | NEU4 | 2:241809065-241817413 | 0.49 | 6.67 | 2.51E-11 | 1.00 | 0.00 | 2.92 | 3.55E-03 | NA | NA | Y |
| 11 | CCDC66 | 3:56557156-56621820 | 0.50 | -4.69 | 2.75E-06 | 0.08 | 0.92 | -3.12 | 1.82E-03 | NA | NA | Y |
| 12 | EAF2 | 3:121835187-121886526 | 0.10 | -5.16 | 2.48E-07 | 0.50 | 0.50 | -2.73 | 6.41E-03 | NA | NA | Y |
| 13 | SLC15A2 | 3:121894324-121944187 | 0.46 | 5.61 | 2.02E-08 | 0.15 | 0.85 | 2.89 | 3.80E-03 | NA | NA | Y |
| 14 | RUVBL1 | 3:128080957-128123828 | 0.29 | -5.18 | 2.22E-07 | 0.97 | 0.03 | -2.28 | 2.26E-02 | NA | NA | Y |
| 15 | TIPARP | 3:156674416-156706768 | 0.08 | 5.08 | 3.72E-07 | 0.15 | 0.84 | 2.56 | 1.05E-02 | NA | NA | Y |
| 16 | FAM114A1 | 4:38867733-38945744 | 0.13 | -5.58 | 2.34E-08 | 0.37 | 0.06 | 1.39 | 1.64E-01 | NA | NA | N |
| 17 | NFKB1 | 4:102501329-102617302 | 0.40 | -4.93 | 8.35E-07 | 0.21 | 0.79 | -1.37 | 1.71E-01 | NA | NA | Y |
| 18 | IL7R | 5:35856875-35879603 | 0.07 | -4.95 | 7.49E-07 | 0.17 | 0.69 | -2.81 | 4.99E-03 | NA | NA | Y |
| 19 | LOC553103 | 5:132311276-132369916 | 0.43 | -4.76 | 1.91E-06 | 1.00 | 0.00 | -3.61 | 3.12E-04 | NA | NA | Y |
| 20 | SLC22A5 | 5:132369709-132395614 | 0.37 | 5.03 | 4.86E-07 | 1.00 | 0.00 | 3.16 | 1.60E-03 | NA | NA | N |
| 21 | SEPT8 | 5:132750817-132777869 | 0.12 | 6.24 | 4.43E-10 | 1.00 | 0.00 | 4.11 | 3.90E-05 | NA | NA | N |
| 22 | RGS14 | 5:177357843-177372598 | 0.50 | -4.89 | 1.02E-06 | 0.26 | 0.73 | -2.96 | 3.05E-03 | NA | NA | N |
| 23 | ITGB8 | 7:20331102-20415759 | 0.08 | -5.73 | 1.02E-08 | 0.64 | 0.29 | -3.81 | 1.40E-04 | NA | NA | Y |
| 24 | JAZF1 | 7:27830574-28180818 | 0.19 | 4.89 | 1.02E-06 | 1.00 | 0.00 | 2.92 | 3.45E-03 | NA | NA | Y |
| 25 | ERMP1 | 9:5784572-5833081 | 0.38 | 5.80 | 6.67E-09 | 1.00 | 0.00 | 4.23 | 2.35E-05 | NA | NA | Y |
| 26 | HHEX | 10:92689924-92695651 | 0.25 | 6.98 | 2.90E-12 | 0.02 | 0.98 | 4.60 | 4.19E-06 | NA | NA | Y |
| 27 | WBP1L | 10:102743970-102816264 | 0.40 | 4.80 | 1.55E-06 | 1.00 | 0.00 | 0.96 | 3.37E-01 | NA | NA | Y |
| 28 | SYT13 | 11:45240302-45286333 | 0.45 | -4.86 | 1.15E-06 | 0.03 | 0.97 | -2.88 | 3.99E-03 | NA | NA | Y |
| 29 | MYO7A | 11:77128264-77215241 | 0.12 | 5.18 | 2.21E-07 | 0.99 | 0.00 | 3.08 | 2.09E-03 | NA | NA | Y |
| 30 | AAMDC | 11:77821162-77872352 | 0.05 | -5.00 | 5.83E-07 | 0.26 | 0.56 | -1.06 | 2.87E-01 | NA | NA | Y |
| 31 | RPS26 | 12:56041902-56044223 | 0.33 | 6.87 | 6.58E-12 | 1.00 | 0.00 | 4.59 | 4.54E-06 | NA | NA | N |
| 32 | NABP2 | 12:56224341-56229854 | 0.07 | -5.99 | 2.06E-09 | 0.63 | 0.03 | -4.45 | 8.69E-06 | NA | NA | Y |
| 33 | SPPL3 | 12:120762510-120904352 | 0.19 | 7.54 | 4.86E-14 | 0.97 | 0.03 | 2.79 | 5.27E-03 | NA | NA | N |
| 34 | ABCB9 | 12:122920951-122966509 | 0.14 | -4.74 | 2.18E-06 | 0.96 | 0.05 | -0.76 | 4.45E-01 | NA | NA | N |
| 35 | CDK2AP1 | 12:123260970-123272316 | 0.44 | 5.26 | 1.41E-07 | 0.48 | 0.52 | 0.86 | 3.87E-01 | NA | NA | N |
| 36 | BAHD1 | 15:40439721-40468242 | 0.05 | 4.74 | 2.19E-06 | 0.62 | 0.08 | 0.83 | 4.07E-01 | NA | NA | Y |
| 37 | CHP1 | 15:41231239-41281885 | 0.33 | -6.04 | 1.53E-09 | 0.79 | 0.21 | -2.73 | 6.26E-03 | NA | NA | N |
| 38 | OIP5-AS1 | 15:41284003-41299597 | 0.19 | -4.69 | 2.70E-06 | 0.87 | 0.13 | -1.35 | 1.78E-01 | NA | NA | N |
| 39 | ITPKA | 15:41493858-41503559 | 0.21 | 5.36 | 8.54E-08 | 0.01 | 0.99 | 2.83 | 4.72E-03 | NA | NA | Y |
| 40 | IQGAP1 | 15:90388241-90502243 | 0.41 | -5.27 | 1.34E-07 | 0.01 | 0.99 | -1.10 | 2.72E-01 | NA | NA | N |
| 41 | DEXI | 16:10928891-10942400 | 0.32 | -7.41 | 1.23E-13 | 1.00 | 0.00 | -2.91 | 3.66E-03 | NA | NA | Y |
| 42 | NSMCE1 | 16:27224994-27268792 | 0.26 | 7.19 | 6.65E-13 | 1.00 | 0.00 | 3.11 | 1.84E-03 | NA | NA | N |
| 43 | IL4R | 16:27313909-27364778 | 0.17 | 5.36 | 8.17E-08 | 0.66 | 0.33 | 2.59 | 9.49E-03 | NA | NA | N |
| 44 | ATP2A1 | 16:28878488-28904509 | 0.08 | -4.62 | 3.80E-06 | 0.30 | 0.66 | -0.97 | 3.33E-01 | NA | NA | Y |
| 45 | LINC00672 | 17:38925168-38929384 | 0.10 | 6.78 | 1.19E-11 | 0.63 | 0.01 | -1.31 | 1.89E-01 | NA | NA | Y |
| 46 | MED1 | 17:39404285-39451274 | 0.17 | 8.20 | 2.46E-16 | 1.00 | 0.00 | -0.11 | 9.16E-01 | NA | NA | N |
| 47 | STARD3 | 17:39637080-39664201 | 0.35 | 14.75 | 3.30E-49 | 1.00 | 0.00 | 2.05 | 4.07E-02 | NA | NA | N |
| 48 | PGAP3 | 17:39671122-39688070 | 0.61 | 15.28 | 9.73E-53 | 1.00 | 0.00 | 1.54 | 1.24E-01 | NA | NA | N |
| 49 | ERBB2 | 17:39688084-39728662 | 0.32 | 13.38 | 7.45E-41 | 1.00 | 0.00 | 0.48 | 6.33E-01 | NA | NA | N |
| 50 | GRB7 | 17:39737909-39747285 | 0.09 | 10.14 | 3.85E-24 | 0.99 | 0.00 | -2.49 | 1.29E-02 | NA | NA | N |
| 51 | IKZF3 | 17:39757715-39864188 | 0.11 | -18.26 | 1.70E-74 | 0.42 | 0.58 | -4.54 | 5.66E-06 | NA | NA | Y |
| 52 | GSDMB | 17:39904595-39918650 | 0.52 | 23.68 | 5.47E-124 | 0.01 | 0.99 | 2.71 | 6.70E-03 | NA | NA | N |
| 53 | GSDMA | 17:39962973-39977766 | 0.34 | -13.40 | 5.74E-41 | 1.00 | 0.00 | -2.93 | 3.43E-03 | NA | NA | N |
| 54 | PSMD3 | 17:39980768-39997960 | 0.06 | 11.33 | 9.54E-30 | 1.00 | 0.00 | 2.18 | 2.95E-02 | NA | NA | Y |
| 55 | MED24 | 17:40019097-40054636 | 0.45 | 5.46 | 4.73E-08 | 1.00 | 0.00 | 0.02 | 9.85E-01 | NA | NA | N |
| 56 | RARA-AS1 | 17:40340867-40343136 | 0.06 | 5.84 | 5.11E-09 | 0.98 | 0.01 | 1.63 | 1.04E-01 | NA | NA | Y |
| 57 | STAT5A | 17:42287547-42311942 | 0.13 | -4.69 | 2.76E-06 | 0.72 | 0.28 | -3.43 | 5.95E-04 | NA | NA | Y |
| 58 | HEXIM1 | 17:45147317-45152101 | 0.27 | -6.01 | 1.80E-09 | 1.00 | 0.00 | -1.24 | 2.14E-01 | NA | NA | Y |
| 59 | SPATA32 | 17:45254393-45262112 | 0.73 | -5.21 | 1.88E-07 | 0.98 | 0.02 | -2.71 | 6.67E-03 | NA | NA | N |
| 60 | LRRC37A4P | 17:45505883-45520523 | 0.32 | 5.66 | 1.55E-08 | 1.00 | 0.00 | 0.17 | 8.64E-01 | NA | NA | N |
| 61 | LOC644172 | 17:45600103-45601862 | 0.10 | -5.71 | 1.16E-08 | 1.00 | 0.00 | -0.46 | 6.44E-01 | NA | NA | Y |
| 62 | MGC57346 | 17:45620329-45637963 | 0.22 | -5.52 | 3.34E-08 | 1.00 | 0.00 | -0.39 | 6.97E-01 | NA | NA | Y |
| 63 | CRHR1-IT1 | 17:45638975-45646229 | 0.26 | -5.63 | 1.82E-08 | 1.00 | 0.00 | -0.42 | 6.78E-01 | NA | NA | N |
| 64 | MAPT | 17:45894382-46028333 | 0.11 | 4.84 | 1.32E-06 | 1.00 | 0.00 | -0.09 | 9.29E-01 | NA | NA | N |
| 65 | KANSL1 | 17:46029916-46225374 | 0.14 | -5.19 | 2.06E-07 | 1.00 | 0.00 | -0.18 | 8.55E-01 | NA | NA | N |
| 66 | KANSL1-AS1 | 17:46193573-46196723 | 0.29 | -5.22 | 1.75E-07 | 1.00 | 0.00 | -0.07 | 9.47E-01 | NA | NA | N |
| 67 | LRRC37A | 17:46295103-46337794 | 0.40 | -6.01 | 1.85E-09 | 1.00 | 0.01 | -0.16 | 8.72E-01 | NA | NA | N |
| 68 | ZNF652 | 17:49289206-49362473 | 0.46 | -5.72 | 1.07E-08 | 0.97 | 0.03 | -3.31 | 9.29E-04 | NA | NA | Y |
| 69 | SPOP | 17:49598884-49678163 | 0.08 | 5.03 | 4.87E-07 | 0.60 | 0.02 | 4.09 | 4.25E-05 | NA | NA | Y |
| 70 | SMAD4 | 18:51030213-51085041 | 0.25 | -6.63 | 3.43E-11 | 0.06 | 0.94 | -2.50 | 1.25E-02 | NA | NA | N |
| 71 | POLI | 18:54269479-54298234 | 0.52 | 6.28 | 3.49E-10 | 0.41 | 0.60 | 3.07 | 2.17E-03 | NA | NA | N |
| 72 | MUC16 | 19:8848844-8981342 | 0.25 | -6.66 | 2.69E-11 | 0.05 | 0.96 | -2.79 | 5.53E-03 | NA | NA | N |
| 73 | FOXA3 | 19:45864260-45873797 | 0.42 | 5.12 | 3.13E-07 | 0.03 | 0.97 | 4.24 | 2.22E-05 | NA | NA | Y |
| 74 | STMN3 | 20:63639705-63653610 | 0.23 | -5.23 | 1.65E-07 | 0.03 | 0.97 | -3.36 | 7.75E-04 | NA | NA | N |
| 75 | LIME1 | 20:63735701-63739107 | 0.18 | 4.74 | 2.14E-06 | 0.72 | 0.28 | 3.00 | 2.66E-03 | NA | NA | Y |
| 76 | RANGAP1 | 22:41244777-41286251 | 0.21 | 5.20 | 2.00E-07 | 0.58 | 0.42 | 2.59 | 9.63E-03 | NA | NA | N |

**Table 1 (continued)**

| No. | Gene | Position (Hg38) | HSQ | Childhood Onset Asthma | | | | Adult Onset Asthma | | | | Nasal Specific? |
|---|---|---|---|---|---|---|---|---|---|---|---|---|
| | | | | TWAS.Z | TWAS.P | PP3 | PP4 | TWAS.Z | TWAS.P | PP3 | PP4 | |
| 77 | TOB2 | 22:41433488-41447023 | 0.05 | -6.02 | 1.74E-09 | 0.05 | 0.95 | -3.87 | 1.09E-04 | NA | NA | Y |
| 78 | PHF5A | 22:41459717-41468704 | 0.05 | 5.92 | 3.22E-09 | 0.27 | 0.73 | 4.15 | 3.34E-05 | NA | NA | Y |
| 79 | XRCC6 | 22:41621163-41664048 | 0.04 | -5.03 | 4.80E-07 | 0.34 | 0.59 | -2.83 | 4.60E-03 | NA | NA | N |
| 80 | C22orf46 | 22:41690543-41698136 | 0.13 | -4.78 | 1.76E-06 | 0.84 | 0.16 | -2.65 | 7.96E-03 | NA | NA | Y |
| 81 | MEI1 | 22:41699514-41799455 | 0.16 | 4.66 | 3.20E-06 | 0.86 | 0.14 | 2.80 | 5.19E-03 | NA | NA | N |
| **Shared** | | | | | | | | | | | | |
| 82 | IL18R1 | 2:102356283-102398775 | 0.08 | -9.16 | 5.05E-20 | 0.54 | 0.05 | -5.31 | 1.11E-07 | 0.56 | 0.02 | N |
| 83 | D2HGDH | 2:241734579-241768816 | 0.64 | 10.31 | 6.63E-25 | 1.00 | 0.00 | 6.61 | 3.85E-11 | 1.00 | 0.00 | N |
| 84 | GAL3ST2 | 2:241776825-241804287 | 0.43 | 8.87 | 7.02E-19 | 1.00 | 0.00 | 6.83 | 8.41E-12 | 1.00 | 0.00 | Y |
| 85 | TSLP | 5:111070080-111078024 | 0.34 | 14.68 | 8.88E-49 | 1.00 | 0.00 | 10.29 | 7.77E-25 | 1.00 | 0.00 | N |
| 86 | PDLIM4 | 5:132257658-132273454 | 0.17 | 7.17 | 7.77E-13 | 1.00 | 0.00 | 7.97 | 1.64E-15 | 0.99 | 0.01 | N |
| 87 | MIR3936 | 5:132365490-132365599 | 0.09 | -5.74 | 9.74E-09 | 1.00 | 0.00 | -4.98 | 6.52E-07 | 1.00 | 0.00 | Y |
| 88 | RAD50 | 5:132556924-132644621 | 0.12 | -8.41 | 4.00E-17 | 0.83 | 0.02 | -7.74 | 9.91E-15 | 0.10 | 0.89 | N |
| 89 | BACH2 | 6:89926528-90296908 | 0.09 | 5.73 | 1.01E-08 | 0.14 | 0.84 | 5.15 | 2.61E-07 | 0.14 | 0.84 | Y |
| 90 | IL33 | 9:6215786-6257983 | 0.25 | 18.84 | 3.67E-79 | 0.00 | 1.00 | 12.41 | 2.18E-35 | 0.01 | 0.99 | Y |
| 91 | RAB5B | 12:55973913-55996683 | 0.05 | 6.23 | 4.57E-10 | 0.42 | 0.15 | 6.25 | 4.22E-10 | 0.07 | 0.86 | Y |
| 92 | UBAC2-AS1 | 13:99196374-99200757 | 0.09 | 5.96 | 2.56E-09 | 0.14 | 0.86 | 5.51 | 3.62E-08 | 0.28 | 0.72 | Y |
| 93 | UBAC2 | 13:99200425-99386499 | 0.29 | -6.60 | 4.20E-11 | 0.04 | 0.96 | -4.98 | 6.32E-07 | 0.20 | 0.80 | Y |
| **Adult Onset Asthma** | | | | | | | | | | | | |
| 94 | RERE | 1:8352404-8817640 | 0.14 | 4.30 | 1.71E-05 | NA | NA | 4.91 | 8.90E-07 | 0.02 | 0.98 | N |
| 95 | ZDHHC18 | 1:26826710-26855720 | 0.07 | 1.27 | 2.04E-01 | NA | NA | 4.62 | 3.85E-06 | 0.09 | 0.86 | N |
| 96 | LOC100130476 | 6:137823670-137868233 | 0.08 | 2.60 | 9.38E-03 | NA | NA | 4.67 | 3.07E-06 | 0.29 | 0.70 | Y |
| 97 | C7orf26 | 7:6590127-6608726 | 0.06 | 2.03 | 4.20E-02 | NA | NA | 4.96 | 7.05E-07 | 0.23 | 0.47 | Y |
| 98 | MUC2 | 11:1074875-1110508 | 0.21 | -2.10 | 3.56E-02 | NA | NA | -5.24 | 1.65E-07 | 0.02 | 0.98 | Y |
| 99 | MUC5AC | 11:1157953-1201141 | 0.32 | 2.74 | 6.21E-03 | NA | NA | 7.22 | 5.13E-13 | 0.02 | 0.98 | Y |
| 100 | FADS1 | 11:61799625-61817057 | 0.32 | 1.62 | 1.05E-01 | NA | NA | 6.49 | 8.49E-11 | 0.04 | 0.96 | Y |
| 101 | SIK2 | 11:111602391-111726917 | 0.27 | -2.63 | 8.44E-03 | NA | NA | -5.06 | 4.10E-07 | 0.21 | 0.79 | N |
| 102 | HDAC7 | 12:47782711-47819980 | 0.16 | -0.98 | 3.29E-01 | NA | NA | -4.62 | 3.92E-06 | 0.39 | 0.60 | Y |

Comparing overlap in nasal TWAS genes with the other TWAS tissues, we found that 52 of the 93 COA and 13 of the 21 AOA nasal TWAS genes were only identified in that tissue (Fig. 2b, c), reflecting the potentially unique transcriptional and pathobiological role of the nasal epithelium in asthma. Furthermore, there were over twice as many nasal-specific COA TWAS genes as there were tissue-specific COA TWAS genes found in any other tissue, including the lung. Similarly, the nasal epithelium yielded the most unique TWAS genes among the different AOA TWAS analyses (Fig. 2b, c).

One of the nasal-specific TWAS genes, interleukin-33 (IL33), for which increased expression was associated with increased asthma risk, was the most strongly associated AOA TWAS gene and the second most strongly associated TWAS gene for COA. This result is consistent with murine models of asthma showing that IL-33 is an epithelial cytokine that initiates T2 airway inflammation cascades that lead to asthma[17,18]. We also observed strong association between increased expression of another epithelial T2 cytokine, TSLP, and risk for both COA and AOA. These results may reflect the prominent role of genetically determined increases in airway epithelial T2 cytokine production on development of both COA and AOA.

The 17q21 locus is of high interest, both because it is the strongest risk locus for COA (based on most GWAS studies including that of Ferreira et al.[6]) and because it is not associated with risk of AOA disease. Previously, 17q21 risk variants have been reported as eQTLs for immune cell expression of ORMDL3 and epithelial expression of GSDMB, and biological experiments support causal mechanisms for these effects[4,5,19–23]. Our nasal TWAS analysis identified 12 COA-associated genes at the 17q21 locus, including 6 that were observed only in the nasal TWAS (LINC00672, MED1, GRB7, IKZF3, PSMD3, RARA-AS1). The strongest nasal TWAS gene association with COA was for GSDMB ($p = 5.5\text{e–}124$), which also showed strong colocalization with the GWAS signal (PP4 = 0.99). Interestingly, GSDMB expression was associated in COA TWAS analyses for all GTEx tissues tested. In contrast, the IKZF3 gene, which exhibited the

third strongest nasal COA association ($p = 1.7\text{e}-74$), was only detected in the nasal analysis (Fig. 3a). This result is especially interesting considering that IKZF3 is an immune cell transcription factor gene[24–26], suggesting that this genetic effect is originating from resident airway immune cells that are captured by our nasal brushings. Although these cell types are most prevalent in the blood, the striking lack of signal in that tissue likely relates to differences in activation and disease phenotype for immune cells when present in the blood versus in a mucosal tissue. In contrast, while we did not observe a nasal eQTL (and thus TWAS association) for the ORMDL3 gene, TWAS analysis of the blood yielded a strong association between ORMDL3 and COA status ($p = 3.9\text{e}-106$, Fig. 3a, b). These analyses support the potential for multiple tissue-specific effects of the 17q21 locus on asthma pathobiology.

**A MUC5AC asthma risk eQTL exerts trans-effects on mucus secretory cell genes.** Intriguingly, we found that two of the TWAS genes that were unique to the nasal airway AOA analysis encode MUC2 ($p = 1.7 \times 10^{-7}$) and MUC5AC ($p = 5.1 \times 10^{-13}$), which are among the four gel-forming mucin genes within the 11p15 locus (MUC6, MUC2, MUC5AC, and MUC5B). Moreover, we found that the strongest eQTL locus (marked by rs12788104) driving expression of both MUC2 ($p = 7.5 \times 10^{-17}$) and MUC5AC ($p = 1.8\times 10^{-31}$) was near completely colocalized with the AOA GWAS signal (COLOC PP4 = 0.98, Fig. 4a), lending further evidence that AOA risk may be modified through these mucin genes. The association between increased MUC5AC expression and AOA risk is especially interesting given that MUC5AC is one of the two major gel-forming mucin proteins that compose airway mucus. Moreover, MUC5AC expression is known to be highly induced in T2-high asthmatics by the action of IL-13. We have previously stratified the GALA II cohort into T2-high and T2-low subjects based on nasal gene expression profiles[15], enabling us to examine the additive and interactive effects of both T2 status and MUC5AC eQTL (rs12788104) genotype on MUC5AC expression in the GALA II cohort using

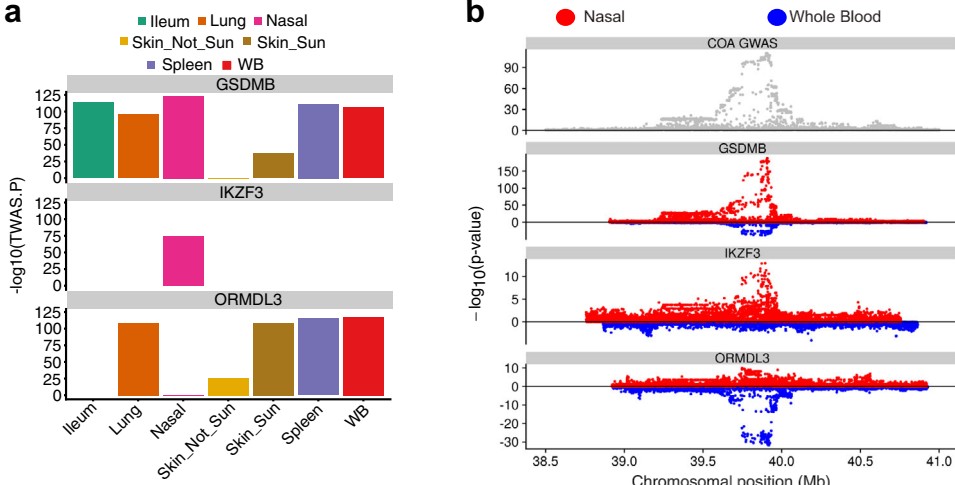

**Fig. 3 Tissue-specific eQTL and TWAS signals at the 17q21 asthma locus. a** Bar plots of different tissue COA TWAS *p* values (two-sided) for *GSDMB*, *IKZF3*, and *ORMDL3*. **b** Local Manhattan plot of COA GWAS on the 17q21 chromosomal region (top). Miami plots comparing whole blood (blue) and nasal epithelium (red) eQTL signals on three 17q21 genes: *GSDMB*, *IKZF3*, and *ORMDL3*. P values were obtained from two-sided tests.

multi-variable modeling (Supplementary Data 5). While this analysis did not support an interaction between these variables ($p = 0.07$), it did reveal rather strong additive effects on *MUC5AC* expression by the rs12788104 variant ($\beta = 0.4$, $p = 4.3 \times 10^{-16}$) and T2 status ($\beta = 0.85$, $p = 6.0 \times 10^{-34}$). In fact, differential expression analysis revealed that the increase in *MUC5AC* expression between rs12788104 GG genotype subjects and AA genotype subjects (FC = 1.63) is equivalent to the increase in *MUC5AC* for T2-high versus T2-low subjects (FC = 1.64). These effects are clearly observed in plotting *MUC5AC* expression stratified by both T2-high status and rs12788104 genotype (Fig. 4b).

IL-13-driven upregulation of *MUC5AC* gene expression leads to increases in both the number of MUC5AC + mucus secretory cells and the amount of MUC5AC protein secreted into airway mucus[12]. This increase in MUC5AC protein secretion is associated with the formation of a tenacious mucus and the arrest of muco-ciliary motion, characteristic of asthma[12]. To evaluate whether rs12788104-induced upregulation of *MUC5AC* expression similarly penetrates to the protein level, we used the air-liquid interface (ALI) culture method to generate mature muco-ciliary airway epithelium organoid cultures from airway stem cells collected from 10 GALA II donors with the rs12788104 AA ($n = 5$) and GG genotypes ($n = 5$). First, based on culture staining for MUC5AC using fluorescent immunohistochemistry, we found a 2.4-fold increase in the number of MUC5AC + secretory cells in GG relative to AA genotype cultures (Fig. 4c, d – left). We next investigated whether this increased abundance of MUC5AC protein within the epithelium also led to increased secretion of MUC5AC protein into mucus by harvesting mucus secretions from all cultures and quantifying MUC5AC protein abundance using an ELISA assay. Strikingly, we found that the GG genotype cultures exhibited a 4.6-fold increase in MUC5AC protein secretion compared to AA cultures (Fig. 4d – right). These in vitro data confirm that rs12788104 is a *MUC5AC* eQTL, while providing evidence that this variant also increases MUC5AC-producing cells and functions as a protein QTL in secreted mucus.

The rs12788104-driven increase in MUC5AC protein production and secretion suggests that the highly specialized ER and Golgi machinery needed to fold, glycosylate, and secrete mucin proteins may be affected by this variant as well. To investigate

this, we performed a transcriptome-wide trans-eQTL analysis for the rs12788104 SNP (Supplementary Data 6). We found that the rs12788104-G asthma risk allele was associated with increased expression of seven genes after correcting for multiple testing. Strikingly, three of these genes (*FKBP11*, *AGR2*, *BHLHA15*) have been demonstrated to play important roles in mucus secretory cell function. Namely, *BHLHA15* (aka MIST1, p-adj = $1.4 \times 10^{-3}$) encodes a master transcription factor that regulates the formation and maintenance of secretory cell architecture[27], *AGR2* (p-adj = $1.3 \times 10^{-2}$) encodes an ER-resident disulfide isomerase involved in folding MUC2 and MUC5AC proteins, which is necessary for their normal production, and *FKBP11* (p-adj = $3.3 \times 10^{-4}$) encodes another isomerase protein that, though not yet linked to mucin folding specifically, is induced by the transcription factor, XBP1, and the unfolded protein response (UPR)[28,29], a pathway well-known to be activated by mucin production. Examining the 453 nominally significant ($p < 0.05$) upregulated trans-eGenes, we found that they were significantly enriched for multiple pathways critical to mucin production and secretion, including *IRE1-mediated unfolded protein response*, *protein N-linked glycosylation*, and *ER to Golgi vesicle mediated transport*, *Golgi subcompartment*, as well as in marker genes of mucus secretory cells determined from single cell RNA-sequencing (scRNA-seq) data[30] (Fig. 4e, Supplementary Data 7). Therefore, we generated an exploratory gene correlation network, encompassing trans-eQTL genes enriched in these pathways and mucus secretory cells, to reveal connections between these functionally related genes (Fig. 4e). We also tested these upregulated trans-eGenes for enrichment within 54 gene co-expression networks that we previously identified for the GALA II transcriptome dataset[15], which capture distinct functional pathways within the nasal airway epithelium (Supplementary Data 8). We found that these upregulated trans-eGenes were strongly enriched in three networks that were all characterized by mucus secretory cell function based on cell type marker[30] and pathway enrichment analyses. These data suggest that the rs12788104 variant not only increases *MUC5AC* expression, but also activates genes needed for the production and secretion of MUC5AC + mucus from airway secretory cells.

Based on the strength of these findings, we lastly considered whether other mucus-related traits recorded in the UK Biobank,

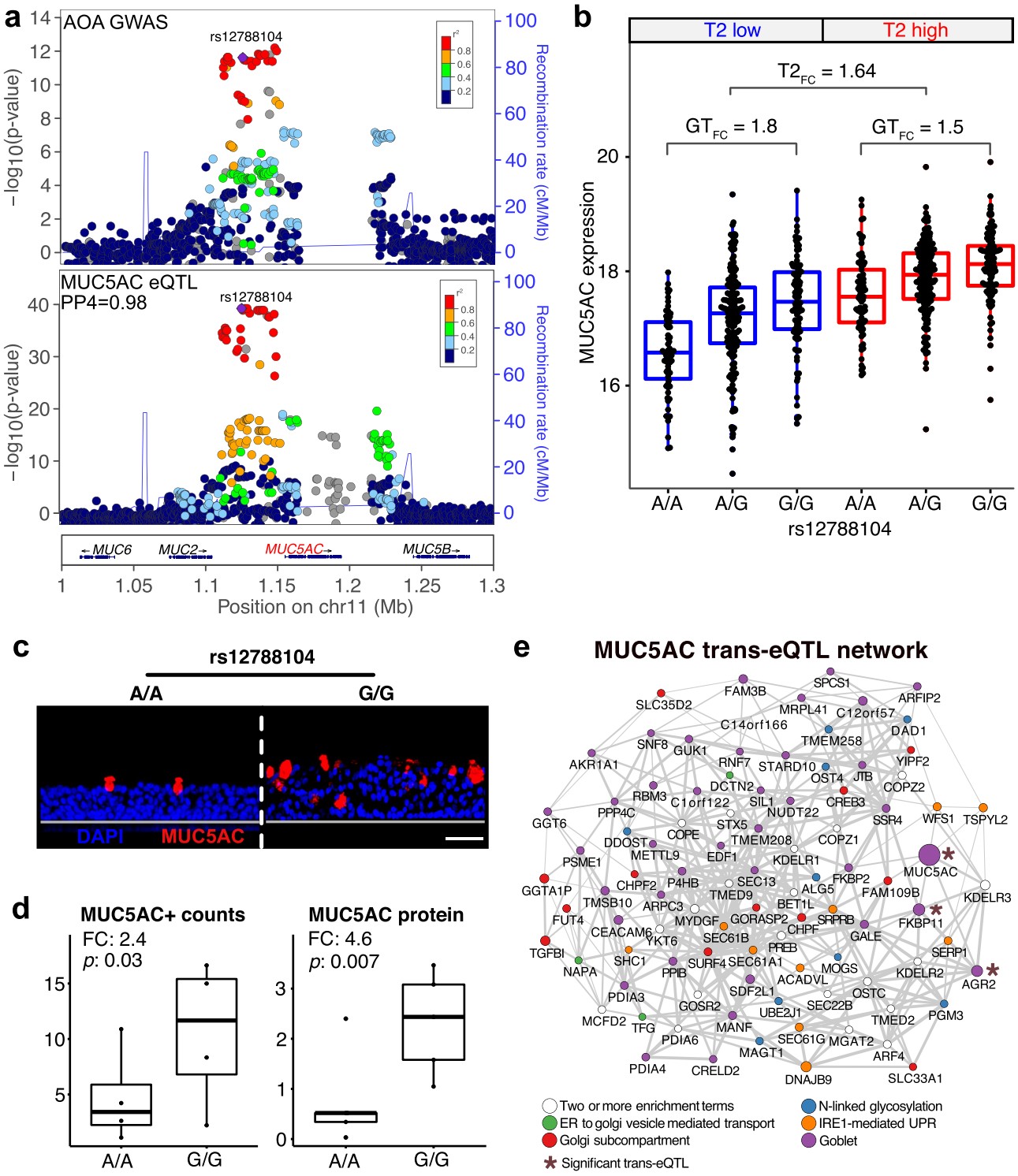

namely *Bring up phlegm/sputum/mucus on most days (22504)* and *Cough on most days (22502)*, both of which are common symptoms of asthma and other mucus obstructive diseases, might similarly be genetically associated with the rs12788104 variant. Based on separate nasal TWAS analyses using these traits, we found that *MUC5AC* was remarkably either one among only seven (for phlegm) or the only (for cough) significantly associated gene(s), with the same rs12788104 marker LD block being positively associated with report of the trait (Supplementary Data 9). These associations with *MUC5AC* suggest that the

rs12788104 variant may not only play a significant role in development of asthma, but may more generally activate mucus production and chronic cough.

**A *FOXA3* asthma risk eQTL drives metaplastic mucus secretory expression.** Also among the nasal-specific TWAS genes (COA $p = 3.1e-7$, AOA $p = 2.2e-5$) was *FOXA3*, a transcription factor whose expression is known to be induced by T2 inflammation, which in turn drives the expression of genes leading to the metaplastic transformation of airway club secretory cells into

**Fig. 4 A *MUC5AC* asthma risk eQTL exerts trans-effects on mucus secretory cell genes. a** Locuszoom plots of AOA GWAS (top) and nasal *MUC5AC* eQTL analyses (bottom) at the gel forming mucin locus on chromosome 11. GWAS *p* values were obtained from two-sided tests. **b** Box plots of normalized *MUC5AC* expression in the GALA II cohort stratified by T2 status and rs12788104 genotype. *N*; T2 low AA: 67; T2 low GA: 166; T2 low GG: 91; T2 high AA: 77; T2 high GA: 180; T2 high GG: 100. Box centers give the median, upper and lower box bounds correspond to first and third quartiles, and the upper/ lower whiskers extend from the upper/lower bounds up to/down from the largest/smallest value, no further than 1.5× IQR from the upper/lower bound (where IQR is the inter-quartile range). Data beyond the end of whiskers are plotted individually. **c** Representative images of MUC5AC IF staining (red) among in vitro ALI nasal airway epithelial cultures from rs12788104 A/A (*n* = 4 donors, 8–11 field of views per donor; left) and G/G (*n* = 4 donors, 8–11 field of views per donor; right) genotype donors. Nuclei were counterstained with DAPI (blue). Scale bars: 50 μm. All images were captured on the Echo Revolve R4 and for each image, brightness and contrasts were uniformly adjusted relative to the brightest feature to balance exposure of each color channel. **d** Box plots of MUC5AC + mean cell counts (left) and secreted MUC5AC mean protein levels (right) between rs12788104 A/A (*n* = 5 donors, 6 technical replicates per donor for MUC5AC protein; *n* = 4 donors, 8–11 field of views per donor for MUC5AC + cell counts) and G/G genotype groups (*n* = 5 donors, 6 technical replicates per donor for MUC5AC protein; *n* = 4 donors, 8–11 field of views per donor for MUC5AC + cell counts). *P* values were obtained from two-sided tests. For MUC5AC + cell count data (left), *p* value was obtained from *GLMMTMB*. For the MUC5AC protein levels data (right), *p* value was provided via Satterthwaite's degrees of freedom method implemented in R package *lmerTest*. Box centers give the median, upper and lower box bounds correspond to first and third quartiles, and the upper/lower whiskers extend from the upper/lower bounds up to/down from the largest/smallest value, no further than 1.5× IQR from the upper/lower bound (where IQR is the inter-quartile range). Data beyond the end of whiskers are plotted individually. **e** Gene correlation network of *MUC5AC* trans-eQTL genes which are enriched among mucus biology-related pathways or among mucus secretory cells. Edges are drawn between each gene and its ten highest correlated genes. Edge thickness: Pearson correlation. Node color: enrichment term. Node size: trans-eQTL LFC estimate. Asterisk next to a node indicates trans-eQTL gene with FDR < 0.05.

specialized, MUC5AC-secreting mucus secretory goblet cells[12,31–33]. The top *FOXA3* eQTL (rs8103278), where the G allele is associated with increased asthma risk, was highly co-localized (PP4 = 0.97) with the *FOXA3* COA GWAS locus (*p* = 1.1e–63, Fig. 5a), strengthening confidence that rs8103278 enhances asthma risk by upregulating this gene.

To examine the relative effects of T2 status (assigned as described above) and the rs8103278 eQTL variant on *FOXA3* nasal expression in vivo, we performed multi-variable modeling using the GALA II cohort. We found that both T2 status and eQTL variant genotype exerted strong additive effects on *FOXA3* expression, although the effect of T2 status was 3.4-fold greater than the genotypic effect (Supplementary Data 10). These effects are clearly observed in plotting *FOXA3* expression by genotype, stratified by T2 status, where T2-low subjects had the lowest *FOXA3* expression, which additively increased by genotype (Fig. 5b). Similarly, *FOXA3* expression also increased additively by eQTL genotype among T2-high subjects, with the T2-high subjects harboring the low-expressing AA genotype, exhibiting expression levels approximately equivalent to T2-low subjects with the high-expressing GG genotype. We next examined whether asthma frequency differed among children in the six strata of *FOXA3* expression created by the intersection of eQTL genotype and T2 status. Interestingly, we found that frequency of asthma was higher among all three T2-high genotype groups than in any T2-low group [asthma OR (T2-high vs. T2-low) = 1.88, *p* = 2e–4] (Supplementary Data 11, Fig. 5c). Strikingly, although we observed no significant genotypic effect on asthma risk in T2-high subjects, we found a strong additive effect of rs8103278 genotype on asthma risk in T2-low subjects (asthma OR (additive genotypic model) = 1.71, *p* = 5.2E–3), suggesting that the effect of *FOXA3* expression on asthma risk may only occur up to a certain threshold of expression, which is commonly exceeded in T2-high subjects. Thus, our results suggest that the greatest impact of the *FOXA3* variant on asthma risk will be among T2-low subjects, where possession of the high expressing genotype can drive *FOXA3* expression levels up to a risk-conferring level, already eclipsed among T2-high subjects.

Considering that FOXA3 is a transcription factor that mediates mucus metaplasia, we next investigated whether the *FOXA3* eQTL variant was associated with expression of any other genes by performing a genome-wide trans-eQTL analysis of the rs8103278 SNP using the GALA II nasal transcriptome (Fig. 5d,

Supplementary Data 12). Interestingly, the top four associated genes (*CLCA1*, *TFF1*, *BPIFB2*, *CLCA2*, *p* < 1e–5), all upregulated in carriers of the G allele (increased *FOXA3* expression), were all well-known markers of IL-13-induced mucus secretory goblet cells. Furthermore, we found that among the co-expression networks previously identified in the GALA II cohort, the larger group of 221 nominally significant (*p* < 0.05) *FOXA3* trans-eGenes were most strongly enriched in a network that contained canonical T2 inflammation response genes, while the second contained mucus secretory genes known to be specifically present in metaplastic T2-induced mucus secretory goblet cells (Fig. 5e, Supplementary Data 8). In contrast, the *FOXA3* trans-eGenes were either not enriched or were only marginally enriched for the three non-inflammatory mucus secretory cell networks that were strongly enriched among *MUC5AC* trans-eGenes (Fig. 5e). These results indicate that genetically driven upregulation in expression of *FOXA3* is sufficient to upregulate expression of genes involved in T2-inflammatory mucus metaplasia. This is in contrast to genetically driven expression of *MUC5AC*, which appears to upregulate genes more broadly involved in mucus production and secretion.

We next investigated whether the genomic localization of the rs8103278 variant was supportive of its strong association with *FOXA3* expression. We found that the rs8103278 variant was within a highly conserved intronic region of the *FOXA3* gene, which has also been experimentally determined by the ENCODE[34] project to be an ELF1 TF binding site (Fig. 5f). These experimental data are reinforced by computational analyses that place the rs8103278 variant at the highly conserved 8th base pair position of the 12 base pair core ELF1 binding motif. In fact, the common G allele of the rs8103278 SNP is present in 96.7% of ELF1 binding motifs. Supporting the likely detrimental effect of the A allele on ELF1 binding, the Ensembl variant effect predictor (VEP)[35] analysis tool found that the A allele of rs8103278 significantly decreased the major allele motif score (Δmotif score = −0.101, Fig. 5g), while Combined Annotation Dependent Depletion (CADD)[36] placed the rs8103278 SNP in the top ~2% of the most deleterious substitutions in the human genome (CADD score = 16.81).

Finally, we investigated whether this genotypic effect on expression was observable in vitro in the context of stimulation with IL-13. We first generated mucociliary airway epithelial cultures from rs8103278 GG (*n* = 5) and AA donors (*n* = 5) and then examined *FOXA3* expression with and without IL-13

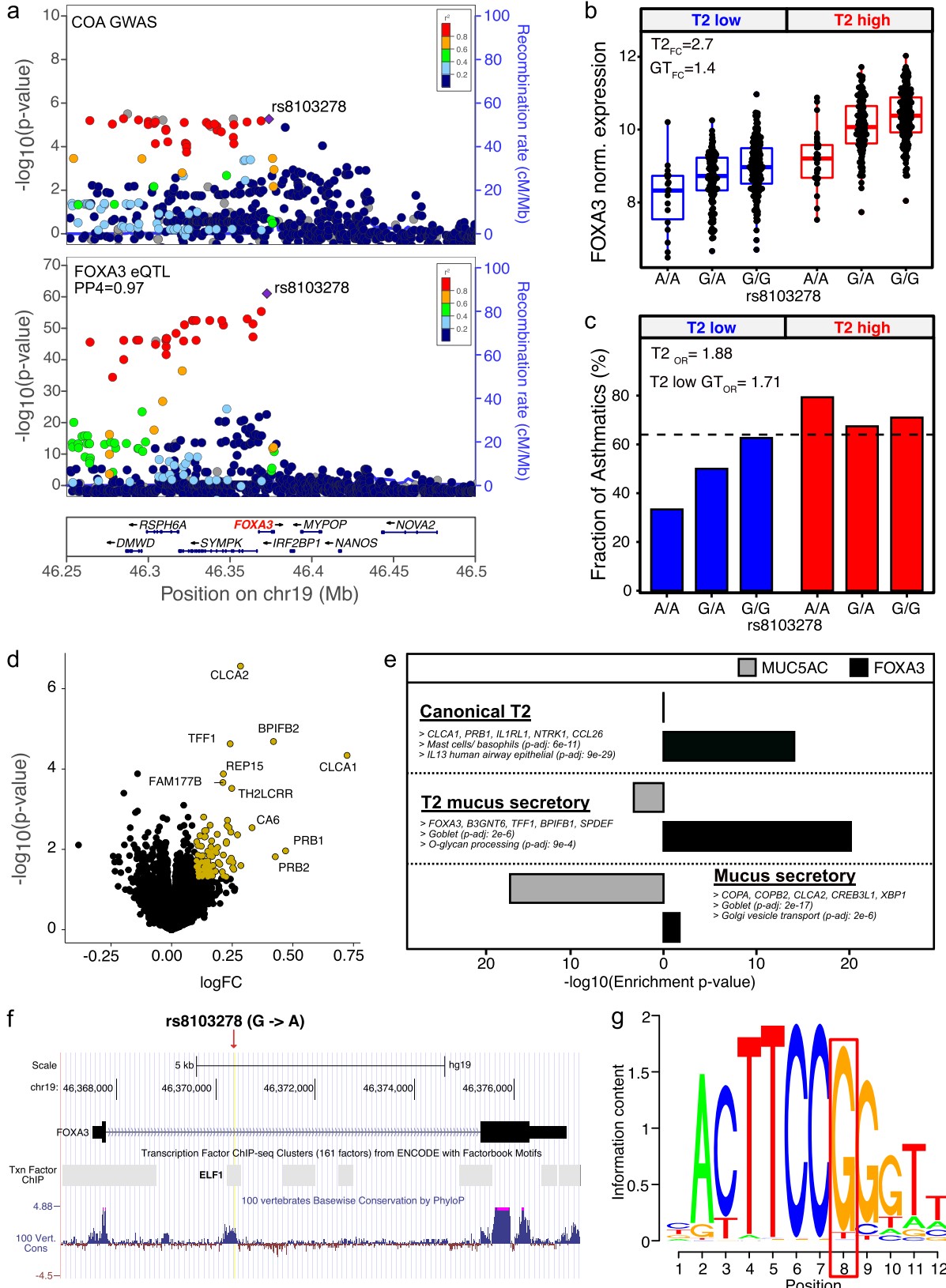

stimulation. As expected, we found that IL-13 stimulation resulted in a dramatic increase in *FOXA3* expression (FC = 52, $p = 6.3e{-}88$, Supplementary Fig. 1). We also found that both IL-13 and control cultures generated from GG donors exhibited significant increases in *FOXA3* expression compared to AA cultures (FC = 2, $p = 3.3e{-}5$, Supplementary Fig. 1), providing further evidence that rs8103278 is the causal SNP at the *FOXA3* TWAS locus.

**Fig. 5 A *FOXA3* asthma risk eQTL drives metaplastic mucus secretory gene expression. a** Locuszoom plots of COA GWAS (top) and nasal *FOXA3* eQTL analyses (bottom) at the *FOXA3* gene locus in chromosome 19. **b** Box plots of *FOXA3* normalized expression in the GALA II cohort stratified by T2 status and rs8103278 genotype. *N*; T2 low AA: 18; T2 low GA: 116; T2 low GG: 190; T2 high AA: 29; T2 high GA: 135; T2 high GG: 193. Box centers give the median, upper and lower box bounds correspond to first and third quartiles, and the upper/lower whiskers extend from the upper/lower bounds up to/down from the largest/smallest value, no further than 1.5× IQR from the upper/lower bound (where IQR is the inter-quartile range). Data beyond the end of whiskers are plotted individually. **c** Bar plots of the percentage GALA II subjects who are asthmatics among the different T2 status by rs8103278 genotype groups. Dashed line indicates the proportion of asthmatics in the full GALA II cohort. *N*; T2 low AA: 18; T2 low GA: 116; T2 low GG: 190; T2 high AA: 29; T2 high GA: 135; T2 high GG: 193. **d** Volcano plot of *FOXA3* (rs8103278) genome-wide trans-eQTL gene analysis. LFC estimate of each gene is obtained from limma by regressing gene expression values on additively coded rs8103278 genotype (A/A: 0; G/A: 1; G/G: 2). Genes with LFC > 0.1 and *p* value (two-sided) < 0.05 are highlighted. **e** Test for enrichment of WGCNA nasal epithelium gene co-expression network (previously identified) genes within *FOXA3* trans-eQTL gene set. The −log10 *p* values for enrichment of network genes within the set of *FOXA3* trans-eQTL genes (black) and *MUC5AC* trans-eQTL genes (gray) are shown with bars. Top network genes and select pathways and cell types enriched among network genes are shown. Enrichment *p* values were obtained from a one-sided hypergeometric test. **f** Genomic location of *FOXA3* eQTL variant (rs8103278) and its colocalization with a ELF1 binding site and cross-species conserved region. **g** ELF1 core binding motif containing the rs8103278 variant. Nucleotide information content is indicated by the size of nucleotide letter abbreviation. The nucleotide in the red box is the rs8103278 SNP site. This high information content position assumes the high-expressing rs8103278 G nucleotide 96.7% of the time, while the alternate rs8103278 A allele is not present.

## Discussion

Here, we have performed the first TWAS analysis for asthma using the nasal airway epithelium, resulting in identification of 114 COA and AOA risk genes. The majority of these genes (*n* = 58), were not identified by TWAS analyses using GTEx consortium expression data from other asthma-related or similar tissues (e.g., lung), suggesting that many of these genes may represent airway-specific aspects of genetically-driven asthma pathobiology. Moreover, our airway TWAS identified at least one significant gene for 33 of the 89 COA risk loci (37%) and 13 of the 40 AOA risk loci (33%) identified by the Ferreira et al. GWAS analysis of UK Biobank data used in our TWAS analysis, affirming the potential importance of airway epithelial dysfunction in asthma development. Importantly, results from these analyses and functional follow-up have revealed a potential genetic basis for two of the most verified and prevalent pathobiological aspects of asthma: T2 inflammation and mucus obstruction. We report airway cis-genetic risk variants for *MUC5AC* and *FOXA3*, alleles of which are associated with both increased expression of these genes and asthma risk. Additionally, these *MUC5AC* and *FOXA3* cis-eQTL risk variants are also associated with trans-effects on the expression of genes contained in non-inflammatory and inflammatory mucus secretory networks, respectively. Moreover, we find that the *MUC5AC* cis-eQTL risk variant is also associated with increased MUC5AC protein secretion into airway mucus, and increased risk of both general chronic cough and mucus production. This study provides some of the first insights into how genetic variants may drive airway inflammation, obstruction, and general airway dysfunction, which in turn contribute to asthma development.

Our study affirms the importance of *tissue context* in identifying genes dysregulated in complex diseases. We found that the nasal airway epithelium exhibited an expression profile that was highly distinct from that of any of the other 26 GTEx tissues, resulting in our identification of many genetically regulated genes that were specific to our airway analysis. In fact, two of the most important findings in this study involved genes (*MUC5AC* and *FOXA3*) that are highly exclusive to mucosal epithelial tissues. Furthermore, these genes are poorly expressed in the airway epithelium when in the absence of environmental exposures (e.g., PM, cigarette smoke) or inflammatory stimuli, which are more likely present among asthmatics[12,32,37,38]. Therefore, we believe that our use of live tissue from a large number of asthmatic subjects aided in our identification of these effects. We acknowledge that the examination of childhood specimens in GALA relative to the older adult specimens in GTEx, may have influenced our results, as differences in gene expression by age

have been reported[39]. We also note that many nasal airway genetic effects are not observed when analyzing lung tissue, despite the fact that lung specimens could contain highly similar bronchial airway epithelia. This result could reflect the difficulty in identifying genetic effects within cellularly diverse organ tissues, likely resulting from low power available for rarer cell types or from conflicting or inconsistent regulation of gene expression across different cell types within the organ. Indeed, a recent comprehensive scRNA-seq analysis of the human lung identified 58 different cell types, of which only 12 are airway epithelial, reflecting the cellular diversity of lung tissue[30]. Moreover, lung explant tissue like that collected by GTEx is mostly parenchymal tissue, dominated by alveolar epithelium, capillaries, macrophage/other immune cells, and fibroblasts, with a rare embedded airway. Additionally, GTEx lung tissue collection protocols directed the collection of distal lung tissue, and to avoid the collection of lung tissue with large bronchi, increasing the likelihood that bronchial airway epithelial cells would be poorly represented among GTEx lung tissue specimens[40]. Another possibility for our unique nasal findings regards differences in statistical power between the nasal and lung analyses, since the GTEx v7 lung sample size was 383 vs. the 681 nasal samples we analyzed. To evaluate this, we compared our nasal TWAS results to a recently published lung/UK Biobank-asthma TWAS[41,42], which used a larger set of 1038 lung donors and >400,000 subject GWAS. This analysis only identified 55 asthma TWAS genes, with only 1 of the 13 AOA and 5 of 52 COA nasal-specific TWAS genes identified among them. These results suggest that power was not a significant reason for the failure to identify these nasal TWAS genes in lung tissue.

The pathobiologic complexity and tissue-specific nature of genetic asthma risk is perhaps best exemplified by our analyses involving the 17q21 COA risk locus. First, while we observed strong COA associations with *GSDMB* in TWAS analysis of both the nasal airway and blood, for *IKZF3* and *ORMDL3*, strong COA associations were only observed in either the airway or blood, respectively, but not both. Secondly, although a subset of 17q21 genes have already been implicated in disparate mechanisms of asthma development (e.g., [1] *ORMDL3* in sphingolipid homeostasis[22,23] and the function of T cells and [2] *GSDMB* in pyroptosis[19,43]), our results suggest additional involvement of the locus based on our identification of 24 COA-associated genes across the six tissue types examined, including 12 from the airway, of which six were airway-specific. Among these airway-specific genes was *IKZF3*, a transcription factor that is not expressed by airway epithelial cells but has been implicated in T, B, and innate lymphoid cell (ILC) development[24–26]. In the

GALA II network analysis, we find that *IKZF3* is part of a cytotoxic T cell network whose expression is highly enriched among GALA II subjects asymptomatically infected with respiratory viruses[15]. This result is especially significant considering that variants in the 17q21 locus have been shown to interact with early-life rhinovirus wheezing illnesses to increase risk of childhood asthma[3]. Thus, given the above, the high COA risk conferred by 17q21 likely involves multiple pathobiologic mechanisms that are driven by several eQTL variants operating in different cell types/tissues.

Outside of the 17q21 locus, variants in *IL33* and *TSLP*, which encode master T2 inflammatory cytokines, are those most strongly associated with COA and AOA. Although previous smaller studies have suggested potential eQTLs for *IL33* and TSLP[44], our eQTL analyses were sufficiently powered to identify strong airway epithelial eQTLs for both genes, which were highly co-localized with asthma GWAS signals. Importantly, that *IL33* eQTL and TWAS signals were only observed in the airway epithelium is consistent with recent studies that implicate the airway epithelium as the primary source of these cytokines in settings of allergic and viral inflammation[7,45,46]. These results strongly suggest that genetically driven upregulation of *IL33/TSLP* in the airway epithelium is a critical mechanism in asthma development.

In asthmatics, a prominent clinical consequence of excessive T2 inflammation is mucus obstruction of the airways, driven by mucus-metaplastic transformation of the epithelium. This transformation entails the appearance of highly specialized mucus secretory goblet cells that use modified pathways of mucin processing and secretion to produce a mucus furnished with distinctive proteins that confer pathologic, visco-elastic properties[12]. Both knockout and transgenic mouse studies, as well as human in vitro studies, have established FOXA3 as an essential transcription factor mediating this T2 mucus-metaplastic process[31,33]. Thus, it is significant that we report here for the first time an airway eQTL for *FOXA3* that confers risk of asthma, which was identified in our nasal TWAS, but not the parent GWAS analysis. The effect of this eQTL was sufficient to drive *FOXA3* expression in T2-low subjects to levels equivalent to those in T2-high subjects, presenting an alternative, genetically driven mechanism for inducing mucus metaplasia in the absence of T2 inflammation. Moreover, we find that this *FOXA3* cis-eQTL exerts trans-expression effects on genes contained in co-expression networks associated with T2-driven mucus metaplasia, suggesting that this genetically driven upregulation of *FOXA3* can itself drive metaplasia of the epithelium. Furthermore, although our results show that asthma risk may be conferred by this locus to T2-low subjects, who would otherwise exhibit low *FOXA3* levels, they suggest that minimal risk is conferred to T2-high subjects, who typically reach risk-conferring levels of *FOXA3* by virtue of T2 inflammation alone. Finally, a causal relationship between rs8103278 and *FOXA3* is suggested by both (1) computational analyses suggesting that the rs8103278 risk allele inhibits binding of ELF1 to the motif in which rs8103278 resides, and (2) our in vitro IL-13 experiments showing that rs8103278 genotype is associated with alteration of *FOXA3* expression. Interestingly, *ELF1* has been reported to be induced by respiratory viral infections[47], providing a potential asthma-related environmental exposure that could activate *FOXA3* expression through this variant.

In addition to this *FOXA3* eQTL, we also identify an eQTL for one of the two major airway gel-forming mucins, *MUC5AC*, which strongly co-localizes with an AOA-specific GWAS locus. The visco-elastic properties of airway mucus are largely determined by the concentration and ratio of MUC5B and MUC5AC mucin proteins, alterations of which are associated with lung and airway diseases[48–50]. MUC5B is generally considered to be the homeostatic mucin, whereas MUC5AC is more inflammatory, being strongly induced by T2 inflammation as well as other asthma-related environmental challenges, including particulate matter and cigarette smoke[32,37,38]. We have found the expression of *MUC5AC* and *MUC5B* to be up- and down-regulated, respectively, in T2-high asthmatics, significantly increasing the *MUC5AC/MUC5B* ratio. Moreover, previous work has found that MUC5AC acts to tether the mucin layer to the epithelial surface, which may contribute to poor muco-ciliary movement[51]. Similar to the effect of rs8103278 on *FOXA3*, we find that the *MUC5AC* eQTL drives *MUC5AC* expression among T2-low subjects to levels equivalent to that of T2-high subjects, suggesting a clinically significant effect. However, unlike the *FOXA3* eQTL, this *MUC5AC* eQTL is associated with trans-effects on genes contained in distinct mucus secretory networks that appear to be unrelated to T2-driven inflammatory processes. Furthermore, this eQTL is associated with increases in both the number of epithelial mucus secretory cells and the amount of MUC5AC protein secreted into mucus. These results indicate that not only does the *MUC5AC* eQTL function as a secreted protein QTL, but that it can possibly stimulate epithelial cells to adopt a mucus secretory phenotype. Remarkably, this same *MUC5AC* eQTL was also associated with both self-reported chronic cough and chronic phlegm production based on TWAS analysis, indicating that this variant-driven mucus production may impose broader effects on respiratory health.

In addition to *FOXA3*, we identified eight other COA and three other AOA genes, not within significant GWAS loci, revealing additional aspects of genetically-driven asthma pathobiology. Interesting among these is the IQ domain GTPase-activating protein (*IQGAP1*) gene, which has been shown to be involved in the proliferation of bronchial airway epithelial cells and to be critical for epithelial repair from injury[52,53]. Another of these, is the Coiled-Coil Domain Containing 66 (*CCDC66*) gene, which is part of centriolar satellites that are involved in cilium assembly[54]. Recently, depletion of CCDC66 from a retinal epithelial cell line was shown to block ciliogenesis[55]. However, no studies have evaluated the potential role of *CCDC66* in airway epithelial ciliogenesis. This information suggests modulation of expression for these genes may mediate asthma risk through a variety of mechanisms, however, further functional studies are needed.

One limitation of this study is that our expression models were generated in the genetically admixed Puerto Rican population and then applied by TWAS analysis to the pre-dominantly European UK Biobank GWAS analyses. Differences in linkage disequilibrium patterns between the eQTL and GWAS populations likely resulted in the loss of some airway TWAS signals, as we have previously demonstrated[56]. Despite this, we were able to identify many clinically significant, co-localized, functionally verified TWAS associations that exhibit strong biological plausibility. Therefore, the airway genetic models developed here should serve as a significant resource for the scientific community, facilitating investigation of the genetic drivers of airway expression that mediate risk for asthma and a range of other lung/airway diseases. We also cannot rule out that some of the uniqueness of the nasal transcriptome we observed could have been driven by methodological differences in RNA-seq data generation between GALA and GTEx. Lastly, although compelling in terms of alignment with our in vivo results and known asthma/airway biology, our in vitro results will need to be confirmed in additional donors, with further investigation of the underlying mechanisms.

In conclusion, our work has established that a significant proportion of genetic risk for asthma may be mediated through

altered gene expression within the airway epithelium. Indeed, we find evidence that these genetically regulated expression effects contribute to the most established pathobiological mechanisms of the disease (through the 17q21 locus, T2 inflammation, and mucus obstruction). Most significantly, we describe two strong cis-regulatory asthma risk variants that drive trans-effects on mucus secretory networks, thus revealing important potential mechanisms of mucus obstruction, that may be therapeutic targets for asthma and other obstructive airway diseases.

## Methods
This research complies with all relevant ethical regulations. The GALA II study protocol was approved by local institutional review boards (NJH HS-2627, University of California, San Francisco [UCSF], IRB number 10–00889, Reference number 153543).

### Experimental methods
*GALA II study subjects.* Nasal airway epithelial RNA-seq data was generated on Puerto Rican children in the Genes-Environment & Admixture in Latino Americans study (GALA II) case-control study of asthma. The GALA II study was approved by local institutional review boards (NJH HS-2627, University of California, San Francisco [UCSF], IRB number 10–00889, Reference number 153543). All GALA II subjects and parents provided written informed assent and written informed consent, respectively[57,58]. The GALA II study and design has been described elsewhere[57–59] and the brief clinical and demographic information are provided here (Supplementary Data 13). In summary, the study recruited Latino asthma and healthy control children (8–21 years), from clinics and community centers in Puerto Rico and the mainland U.S. Asthma was physician-diagnosed, with a requirements for 2 or more symptoms of coughing, wheezing, or shortness of breath within 2 years of study enrollment. Exclusion criteria included: subjects in the third trimester of pregnancy, current smoking or at least a 10 pack-year smoking history. GALA II subjects completed questionnaires detailing demographic, medical, and environmental data. A blood sample was obtained to isolate DNA for Whole Genome Sequencing (WGS). All GALA II subjects used in this analysis were recruited from Puerto Rico (n = 695). A nasal airway inferior turbinate brushing was used to collect airway epithelial cells from GALA II subjects for whole transcriptome sequencing (n = 695). The GALA II WTS data have been used in several prior publications[12,15,60]. Network analyses leveraged here were performed as part of another publication[15]. The eQTL and TWAS analyses were performed on the subset of subjects (n = 681) with WGS and nasal WTS data.

*ALI culture of primary nasal airway epithelial cells.* Nasal airway epithelial brushes were obtained from GALA II study subjects from which basal airway stem cells were cultured for expansion using a modified Schlegel method as previously described[61–63]. Basal stem cells from subjects with rs12788104 AA (n = 5) and GG (n = 5) genotypes and rs8103278 GG (n = 5) and AA (n = 5) genotypes were seeded onto 6.5 mm 24-well polyester Transwell inserts with 0.4 µm pore size at 1.2 × 10^5 cells/cm^2 in ALI Expansion Medium supplemented with Y-27632 as previously described, and were air-lifted upon development of an intact monolayer. Cultures were differentiated using PneumaCult ALI (PC-ALI) differentiation media (StemCell Technologies) for 21 days, then harvested for gene expression, immunofluorescence, and mucin ELISA analysis.

*Bulk RNA sequencing of ALI culture samples.* RNA lysates were harvested from each mature culture and extracted using the Quick-RNA MiniPrep Kit (Zymo Research). RNA Normalization, library construction using KAPA mRNA HyperPrep Kit (KAPA Biosystems/Roche), and library pooling were all performed on the Beckman Coulter FX^P automation system. RNA samples were randomized over the normalization plate at 250 ng per sample, and barcodes were added using 12 cycles of amplification to generate sequencing libraries for whole transcriptome sequencing using the Illumina NovaSeq 6000 with 150 bp paired end reads.

*Bulk RNA sequencing of GALA II samples.* Total RNA was extracted from the nasal airway epithelial brushings of GALA II subjects (n = 695) using the AllPrep DNA/RNA Mini Kit (QIAGEN, Germantown, MD). We used 250 ng of total RNA as input for the KAPA Stranded mRNA-seq library kit (Roche Sequencing and Life Science, Kapa Biosystems, Wilmington, MA), using the Beckman Coulter Biomek FX^P automation system (Beckman Coulter, Fullerton, CA). Barcoded libraries were pooled and sequenced on the Illumina HiSeq 2500 system using 125 bp paired-end reads (Illumina, San Diego, CA).

*Whole genome sequencing of GALA II samples.* Whole blood obtained from GALA II study subjects was used to extract genomic DNA using the Wizard Genomic DNA Purification kits (Promega, Fitchburg, WI). Extracted DNA was quantified by fluorescent assay. Whole genome sequencing of DNA was performed as part of the

Trans-Omics for Precision Medicine (TOPMed) whole genome sequencing (WGS) program[64]. WGS was performed at the New York Genome Center and the Northwest Genomics Center on a HiSeqX system (Illumina, San Diego, CA) using a paired-end read length of 150 base pairs, with a minimum of 30X mean genome coverage. Sequence data were mapped using BWA-MEM[65] (v0.7.15) to the hs38DH 1000 Genomes build 38 human genome reference with the options "-K 100000000 -Y". Variants were identified and called using the GotCloud[66] pipeline (v1.17). Variants were filtered for quality, and genotypes with at least 10X coverage were phased with Eagle 2.4[67]. Variant calls used in this study were obtained from TOPMed data freeze 8 variant call format files.

*MUC5AC ELISA.* Quantification of secreted MUC5AC protein was performed by ELISA as previously described[68,69]. Differentiated mucociliary epithelial ALI cultures were grown for 21 days. Cultures were washed twice with PBS and allowed to rest for 48 h. Culture plates were carefully washed with warm PBS for 30 min, for 4 total washes, to remove accumulated apical mucus. Following washes, 100ul of ATP-y-S (100 mM) was added to the apical surface and incubated for 1 h at 37 °C. Apical suspensions were collected to quantify secreted mucin abundance. For MUC5AC ELISA, collected samples were diluted 1:100 in PBS and 100ul of each sample was added, in duplicate, to a High Bind MaxiSorp microtiter plate and incubated for 2 h at 37 °C. Wells were washed with 1x Phosphate Buffered Saline with 0.05% Tween-20 (PBST), blocked with 5% milk in PBST for 1 h at 37 C, and washed again with PBST. Plates were incubated with mouse anti-MUC5AC (1:1000) in 1% milk/PBST overnight at 4 C, washed with PBST, and incubated with secondary HRP-conjugated anti-mouse IgG (1:1000) for 1 h at 37 °C, followed by a final round of PBST washes. Plates were developed using OPD substrate/Phosphate Citrate Buffer Solution with 10 µl 30% hydrogen peroxide for 15 min at room temperature, and reactions were stopped by addition of 4 M $H_2SO_4$. Plate absorbance was measured at 490 nm on a microtiter plate reader.

*Immunofluorescence labeling.* Intact ALI culture inserts were fixed for 15 min at room temperature in 3.2% paraformaldehyde. Histology sections were stripped of paraffin using HistoChoice Clearing Agent (Sigma), rehydrated using a decreasing gradient of alcohol washes (100%, 90% 70%, 50%, 30%, 0%), and antigen retrieval conducted using Citric Acid-based Antigen Unmasking Solution pH 6.0 (Vector Laboratories). Prior to labeling, all samples were blocked and permeabilized for 30 min using 3% BSA/0.1% Triton X-100 in tris-buffered saline (TBS). Primary labeling was conducted using 3% BSA/0.1% Triton X-100 in TBS for 1 h with mouse anti-MUC5AC (1:500; ThermoScientific). Sections were washed twice using TBS/0.1% Triton X-100 (TBST) and secondary labeling was conducted using DAPI (1:1000) and AlexFluor594 goat anti-mouse IgG (1:500; ThermoScientific) for 30 min. Slides were washed twice in TBST and mounted with Vectashield HardSet Antifade Mounting Media (Vector Laboratories). Images were acquired using a Revolve microscope (Echo Labs) at 10x magnification.

### Quantification and statistical analysis
*Analysis of GALA II RNA-seq data.* The preprocessing steps of GALA II RNA-seq data analysis, including read trimming (skewer v0.2.2[70]; end-quality=15, mean-quality=25, min=30), aligning of reads to the human genome (GSNAP v20160501; max-mismatches=0.05, indel-penalty=2, batch=3, expand-offsets=0, use-sarray=0, merge-distant-same-chr), and gene quantification (htseq-count v0.9.1)[71] have been previously described[15]. The raw count matrix was then variance stabilized using DESeq2[72] (v1.22.2). Network analysis was carried out on the variance-stabilized count matrix to identify co-expressed genes networks with WGCNA[73] (v1.68; soft threshold=9, minClusterSize=20, deepSplit=2, pamStage=TRUE) and T2 endotype assignment based on the hierarchical clustering of T2 module genes was performed previously[15]. Briefly, we hierarchically clustered all subjects based on expression of only the genes in the T2 network (Supplementary Data 14) and then used the first split in the dendrogram as the basis for assigning individuals to T2-high or T2-low categories.

*Analysis of combined GTEx and GALA II RNA-seq data.* The GTEx v7 raw gene expression count matrix was downloaded from the GTEx[16] portal (https://gtexportal.org/) and combined with the GALA II raw count matrix. The combined GALA II and GTEx raw gene expression count matrix containing the 22,515 common genes was then trimmed mean of M-values (TMM) normalized using edgeR[74] (v3.22.3). Multidimensional scaling (MDS) analysis was then performed on the top 10,000 most variable genes from the TMM normalized expression matrix (Fig. 1a). Tissues hierarchical clustering was performed on the median gene expression levels using Pearson correlation as the distance metric and using the average linkage method (Fig. 1b).

*Cis-eQTL analysis.* Cis-expression quantitative trait locus (eQTL) analysis of the nasal RNA-seq data and WGS variant data from 681 GALA II subjects has been described in detail elsewhere[15]. Briefly, cis-eQTL analysis on 12,590,800 genetic variants and 17,039 genes was performed by following the general methodology of the GTEX project version 7 protocol using a MAF cutoff of 1% and a cis window of 1 Mb using FastQTL (v2.184)[75] to identify eGenes and eVariants. Furthermore, we

performed stepwise regression analysis to identify independent eQTL variants using QTLTools[76] (v1.1). ADMIXTURE[77] (v1.3.0) was used to compute admixture factors and PEER[78] (v1.3) was used to generate PEER factors. Heritability estimate for the expression of each gene was computed with GCTA[79] (v1.26.0) using genetic markers within the cis region of the gene.

*Transcriptome-wide association study.* Transcriptome-wide association study (TWAS) analyses were performed by a two-step process using TWAS/FUSION[13] software version 1.0. First, we utilized 681 donors with paired nasal epithelial transcriptomes and genotype data to build gene expression prediction models for 17,039 expressed genes (as defined in the cis-eQTL method section above). For each expressed gene in the nasal epithelium, a gene expression prediction model was built incorporating genotype variants in the cis-locus of the gene (+/− 1MB on either side of the gene boundary) using the FUSION.compute_weights.R script with models blup, top1, and enet, resulting in the creation of expression weights for 12,523 heritable genes (heritability two-sided *p* value < 0.01).

Gene association testing was performed by combining the nasal epithelium gene expression weights and GWAS summary statistics of asthma related traits (i.e.: AOA, COA, cough, and phlegm) using the FUSION.assoc_test.R R script using 1 kg EU LD reference panel and max-impute=0.9. Colocalization[80] statistics (PP4: Posterior Probability of colocalized eQTL/GWAS associations) were computed for genes with TWAS *p* values < 1e−5 (coloc_p = 1e−5). Bonferroni correction was used to adjust for multiple testing within a tissue-trait combination. For TWAS using the nasal epithelium expression model, the *p* value threshold of 3.99e−6 (0.05/12,523 heritable genes) was used to declare significant TWAS genes. For association testing using other asthma relevant tissues from GTEx_v7 (skin: *n* = 335 and *n* = 414, lung: *n* = 383, whole blood: *n* = 368, ileum: *n* = 122, and spleen *n* = 146), we obtained expression weights from the FUSION website (http://gusevlab.org/projects/fusion/#reference-functional-data). GWAS summary statistics for AOA and COA were obtained from https://genepi.qimr.edu.au/staff/manuelF/gwas_results/main.html. GWAS summary statistics for cough and phlegm were obtained from http://www.nealelab.is/uk-biobank/.

*Analysis of RNA-seq, cell counts, and protein data from the nasal cultures.* For RNA-seq data, raw sequencing reads were trimmed using skewer[70] with the following parameter settings: end-quality=15, mean-quality=25, min=30. Trimmed reads were then aligned to the human reference genome GRCh38 using HISAT2[81] (v2.1.0) using default parameter settings. Gene quantification was performed with htseq-count[71] using the GRCh38 Ensembl v84 gene transcript model. After removing lowly expressed genes (those not adequately expressed (>5 counts) in at least three samples), we carried out differential expression analyses between the samples with GG (*n* = 5 donors) and AA genotypes (*n* = 5 donors) for *FOXA3* eQTL variant rs8103278 (Supplementary Fig. 1) using the DESeq2[72] R package (v1.22.2).

Protein expression data were fitted with a linear mixed model, implemented in the R package, *lme4*[82] (v1.1.26) using an *lmer* function with parameter – REML = FALSE, to compare the protein expression level between the two genotype groups (rs12788104:A/A; *n* = 5 donors and rs12788104:G/G; *n* = 5 donors, 6 technical replicates per donor) accounting for age and T2 status as fixed effects, and donor ID as random intercept. Two-sided *p* values were provided via Satterthwaite's degrees of freedom method implemented in the R package, *lmerTest* (v3.1.3)[83]. Similarly, MUC5AC + cell counts data (rs12788104:A/A; *n* = 4 donors and rs12788104:G/G; *n* = 4 donors, 8–11 field of views per donor) were fitted with a negative binomial linear mixed model implemented in the R package, *glmmTMB*[84] using the same model as above.

*Trans-eQTL analysis of MUC5AC (rs12788104) and FOXA3 (rs8103278) eQTL variants on GALA II RNA-seq data.* We performed trans-eQTL analysis on *FOXA3* eQTL variant (rs8103278) and *MUC5AC* eQTL variant (rs12788104) using the normalized GALA II RNA-seq data described previously (see cis-eQTL analysis section above). Limma (v3.46)[85] was used to fit a regression model with normalized gene expression as the outcome variable and additively coded SNP genotype as the main predictor and adjusting for T2 status, age, sex, BMI, and admixture estimates. *P* values were obtained from two- sided tests.

*Multivariable regression analysis of MUC5AC, FOXA3, and asthma status.* We performed multiple regression analysis using R's *lm* function to evaluate the association of both *MUC5AC* and *FOXA3* expression with its eQTL variant and T2 status, controlling for the following covariates: age, sex, BMI, asthma status, and admixture variables (Supplementary Data 5 and 9, two-sided *p* values are reported). We fitted the model with and without the interaction term between T2 status and the genotype of the eQTL variant (rs12788104 for *MUC5AC* and rs8103278 for *FOXA3*).

Similarly, we fitted a logistic regression model using R's *glm* function on asthma status using the following predictors: age, sex, BMI, admixture variables, rs8103278, T2 status, and their interactions (Supplementary Data 11, two-sided *p* values are reported).

*Gene set enrichment analysis.* To investigate enriched pathways from trans-eQTL genes or WGCNA network genes (see Figs. 4e and 5e), we used enrichrR[86] (v3.0) to test for overrepresentation of network genes within a panel of annotated gene databases (Gene Ontology [GO] Biological Process [BP] 2018, GO Molecular Function [MF] 2018, GO Cellular Component [CC] 2018, Ligand Perturbations from GEO up, Kyoto Encyclopedia of Genes and Genomes [KEGG] 2019 Human, and Reactome 2016). Additionally, gene marker sets were obtained for each of 35 epithelial and immune cell types inferred from a recent scRNA-seq study on ~70,000 cells from human lung tissue samples obtained intraoperationally from three individuals[30]. Similarly, the trans-eQTL and WGCNA network genes were tested for overrepresentation within each of these marker sets.

*Variant annotation and motif analysis for rs8103278.* The CADD[36] v1.5 database was downloaded from the CADD database website (https://cadd.gs.washington.edu/) and the CADD score for rs8103278 (19:45867123:G:A) was extracted. The Ensembl variant effect predictor (VEP)[35] tool was run to find the list of possible transcription factor binding motifs overlapping with rs8103278. Functional annotations on the genomic region surrounding the genetic variants of interest were downloaded from the UCSC genome browser[87] with a specific focus on conservation scores (Phylop[88]) and transcription factor binding sites from ENCODE CHIP-seq data (Fig. 5f). The motif logo for ELF1 (MA0473.2) was downloaded from the JASPAR[89] database using the JASPAR2020[89] R package (v0.99.10). The motif information content matrix (ICM) was computed from a raw frequency matrix using the R package, TFBSTools[90] (v1.28.0) with pseudocounts = 0.8 and schneider=T. The ELF1 motif sequences were reverse complemented prior to display (Fig. 5g).

**Reporting summary.** Further information on research design is available in the Nature Research Reporting Summary linked to this article.

## Data availability

GALA II RNA-seq data used in this study has been previously deposited in the National Center for Biotechnology Information/Gene Expression Omnibus (GEO) GSE152004. The processed protein and RNAseq data used in the invitro experiments are available in the github repository [https://github.com/seiboldlab/Nasal_TWAS/tree/main/Data]. Data originating from public repositories can be accessed at the following location: GO Biological Process 2018 table [https://maayanlab.cloud/Enrichr/geneSetLibrary?mode=text&libraryName=GO_Biological_Process_2018]; GO Molecular Function 2018 table [https://maayanlab.cloud/Enrichr/geneSetLibrary?mode=text&libraryName=GO_Molecular_Function_2018]; GO Cellular Component 2018 table [https://maayanlab.cloud/Enrichr/geneSetLibrary?mode=text&libraryName=GO_Cellular_Component_2018]; Ligand Perturbations from GEO up table [https://maayanlab.cloud/Enrichr/geneSetLibrary?mode=text&libraryName=Ligand_Perturbations_from_GEO_up]; Kyoto Encyclopedia of Genes and Genomes 2019 Human table [https://maayanlab.cloud/Enrichr/geneSetLibrary?mode=text&libraryName=KEGG_2019_Human]; Reactome 2016 table [https://maayanlab.cloud/Enrichr/geneSetLibrary?mode=text&libraryName=Reactome_2016]; TOPMed freeze 8 variant calls are available from dbGaP accession phs000920.v2.p2 [https://www.ncbi.nlm.nih.gov/projects/gap/cgi-bin/study.cgi?study_id=phs000920.v4.p2]; CADD v1.5 database [https://cadd.gs.washington.edu]; JASPAR 2020 database [http://jaspar.genereg.net]; Ferreira et. al. UKBB COA GWAS summary statistics [https://genepi.qimr.edu.au/staff/manuelF/gwas_results/CHILD_ONSET_ASTHMA.20180501.allchr.assoc.GC.gz]; Ferreira et. al. UKBB AOA GWAS summary statistics [https://genepi.qimr.edu.au/staff/manuelF/gwas_results/ADULT1_ADULT2_ONSET_ASTHMA.20180716.allchr.assoc.GC.gz]; UKBB cough (22502) GWAS summary statistics [https://broad-ukb-sumstats-us-east-1.s3.amazonaws.com/round2/additive-tsvs/22502.gwas.imputed_v3.both_sexes.tsv.bgz]; UKBB phlegm (22504) GWAS summary statistics [https://broad-ukb-sumstats-us-east-1.s3.amazonaws.com/round2/additive-tsvs/22504.gwas.imputed_v3.both_sexes.tsv.bgz].

Further information and requests for resources and reagents should be directed to and will be fulfilled by Max A. Seibold, Ph.D. (seiboldm@njhealth.org)

## Code availability

The scripts used to perform the analyses described in the paper have been deposited in the github repository [https://github.com/seiboldlab/Nasal_TWAS][91].

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

## Acknowledgements
This work was supported by NIH grants (MAS) R01 HL135156, R01 MD010443, R01 HL128439, P01 HL132821, P01 HL107202, R01 HL117004, and DOD Grant W81WH-16-2-0018. The Genes-Environments and Admixture in Latino Americans (GALA II) Study and E.G.B. were supported by the Sandler Family Foundation, the American Asthma Foundation, the RWJF Amos Medical Faculty Development Program, the Harry Wm. and Diana V. Hind Distinguished Professor in Pharmaceutical Sciences II, the National Heart, Lung, and Blood Institute (NHLBI) [R01HL117004, R01HL128439, R01HL135156, X01HL134589]; the National Institute of Environmental Health Sciences [R01ES015794]; the National Institute on Minority Health and Health Disparities (NIMHD) [P60MD006902, R01MD010443], the National Human Genome Research Institute [U01HG009080] and the Tobacco-Related Disease Research Program [24RT-0025, 27IR-0030]. Burchard NIH Support: R01 128439, R01 HL141992, R01 HL141845. Whole genome sequencing (WGS) for the Trans-Omics in Precision Medicine (TOPMed) program was supported by the National Heart, Lung and Blood Institute (NHLBI). WGS for NHLBI TOPMed: Gene-Environment, Admixture and Latino Asthmatics Study (phs000920) was performed at the New York Genome Center (3R01HL117004-02S3) and the University of Washington Northwest Genomics Center (HHSN268201600032I). Centralized read mapping and genotype calling, along with variant quality metrics and filtering were provided by the TOPMed Informatics Research Center (3R01HL-117626-02S1; contract HHSN268201800002I). Phenotype harmonization, data management, sample-identity QC, and general study coordination were provided by the TOPMed Data Coordinating Center (3R01HL-120393-02S1, U01HL-120393, contract HHSN268201800001I). We gratefully acknowledge the studies and participants who provided biological samples and data for TOPMed. WGS of part of GALA II was performed by New York Genome Center under The Centers for Common Disease Genomics of the Genome Sequencing Program (GSP) Grant (UM1 HG008901). The GSP Coordinating Center (U24 HG008956) contributed to cross-program scientific initiatives and provided logistical and general study coordination. GSP is funded by the National Human Genome Research Institute, the National Heart, Lung, and Blood Institute, and the National Eye Institute. We wish to acknowledge the following GALA II study collaborators: Shannon Thyne, UCSF; Harold J. Farber, Texas Children's Hospital; Denise Serebrisky, Jacobi Medical Center; Rajesh Kumar, Lurie Children's Hospital of Chicago; Emerita Brigino-Buenaventura, Kaiser Permanente; Michael A. LeNoir, Bay Area Pediatrics; Kelley Meade, UCSF Benioff Children's Hospital, Oakland; William Rodríguez-Cintrón, VA Hospital, Puerto Rico; Pedro C. Ávila, Northwestern University; Jose R. Rodríguez-Santana, Centro de Neumología Pediátrica; Luisa N. Borrell, City University of New York; Adam Davis, UCSF Benioff Children's Hospital, Oakland; Saunak Sen, University of Tennessee. The authors acknowledge the families and patients for their participation and thank the numerous health care providers and community clinics for their support and participation in GALA II. In particular, the authors thank the recruiters who obtained the data: Duanny Alva, MD; Gaby Ayala-Rodríguez; Lisa Caine, RT; Elizabeth Castellanos; Jaime Colón; Denise DeJesus; Blanca López; Brenda López, MD; Louis Martos; Vivian Medina; Juana Olivo; Mario Peralta; Esther Pomares, MD; Jihan Quraishi; Johanna Rodríguez; Shahdad Saeedi; Dean Soto; and Ana Taveras. The content is solely the responsibility of the authors and does not necessarily represent the official views of the National Institutes of Health.

## Author contributions
Conceptualization: M.A.S., S.P.S.; Formal analysis: S.P.S., B.S.; Methodology: S.P.S., B.S., N.D.J., C.M.M,. M.A.S.; Investigation: S.P.S., J.L.E., C.L.R., A.F.M., C.E., M.A.S.; Visualization: S.P.S., N.D.J., B.S., M.A.S.; Funding acquisition: M.A.S., E.G.B.; Project administration: J.E,. V.M., J.R.S., S.S,. S.O., M.A.S.; Resources: M.A.S., A.C.M., C.E., S.G., J.E., S.H,. D.A.N., S.G., M.C.Z., G.A., H.M.K., K.M.R., R.K., S.O., J.R.S., V.M., E.G.B.; Supervision: M.A.S.; Writing—original draft: S.P.S., M.A.S.; Writing—review & editing: S.P.S., N.D.J., J.L.E., C.M.M., R.K., M.A.S,. N.Z., E.G.B.;. The NHLBI Trans-Omics for Precision Medicine (TOPMed) Consortium authors contributed to the TOPMed Whole Genome Sequencing data collection, joint processing and quality controls, and establishment of analysis procedures.

## Competing interests
H.M.K. is an employee of Regeneron Pharmaceuticals and owns stock and stock options for Regeneron Pharmaceuticals. G.A. is an employee of Regeneron Pharmaceuticals and owns stock and stock options for Regeneron Pharmaceuticals. M.C.Z. owns stock in ThermoFisher and Merck. All other authors declare no competing interests.

## Additional information

## NHLBI Trans-Omics for Precision Medicine (TOPMed) Consortium

Deborah A. Nickerson[6], Soren Germer [7], Michael C. Zody [7], Gonçalo Abecasis [8], Hyun Min Kang[8], Kenneth M. Rice [9] & Esteban G. Burchard[4,12]

A full list of members and their affiliations appears in the Supplementary Information.

