## [Peer Review File · Nature Communications]

Transcriptome-wide association study of asthma, using nasal airway epithelium, reveals genetically-driven mucus pathobiology.Reviewers' Comments:

Reviewer #1:

Remarks to the Author:

This is an interesting study adding significantly to the understanding of the molecular mechanisms of asthma.

The manuscript is well written, and the methodologies seem sound.

Major points

My main concern relates to the interpretation/reason for the finding that nasal gene expression is markedly different from other tissues, including lung tissue, and performs markedly better in the TWAS comparison.

There are several possible explanations for this finding that needs to be analysed and discussed in more details:

As mentioned in the introduction, the authors have previously shown that the nasal airway epithelium expresses the same genes as the bronchial airway, epithelium, with highly correlated expression levels between these sites. This seems contradictory to the finding that nasal gene expression shows no co-clustering with lung expression in GTEX.

This raises the possibility that this could be due to methodological differences, eg due to platform or other differences in generating and handling the "local" and GTEX datasets. Please address this, potentially by including other eQTL sources.

Could the better results using nasal data be due to the younger age of individuals? Please compare with GTEX participant characteristics.

The authors mention the larger sample size in their nasal eQTL data as a strength – how does their nasal sample size compare to the lung sample size in GTEX?

Another important point is that a large proportion of individuals providing nasal eQTL data were asthmatics. This is mentioned shortly in the discussion but not analytically. Probably very few individuals in the GTEX dataset were asthmatics?

First, I wonder if this could lead to some inflation of the eQTL signals as a result of differential expression between asthmatics and controls?

Second, the possibility that the improved performance of the nasal data is due to more relevant eQTL data in asthmatics is important for directing future studies.

The effect of having asthmatics in the nasal eQTL data should therefore be analyzed, eg by stratified analyses, at least in a targeted way focusing on the most prominent TWAS findings.

If the stronger results using nasal data is a true biological finding, does it suggest that the nasal epithelium is more important than the lung in asthma inception?

Regarding the MUC5AC trans-eQTL analysis:

The authors suggest that the MUC5AC eQTL increases asthma risk by not only increasing MUC5AC expression, but also through activating networks of genes needed for the production and secretion of MUC5AC+ mucus from airway secretory cells. However, given the biological property of MUC5AC, as a mucin and not a transcription factor or immune mediator, it seems equally plausible that the trans-effects observed could be responses to altered mucus properties, without any effect on asthma risk. Without any direct evidence that these trans-effects are related to asthma risk, I think a more careful interpretation of these trans-signals is warranted.

Minor point

The authors identified 8 COA TWAS genes and a single AOA TWAS gene that were not within a Mb of any GWAS risk variant. While most of the other TWAS genes have previously been suspected as causal genes underlying asthma loci, these 9 are the most novel potential risk genes and should be discussed.

Reviewer #2:

Remarks to the Author:

Sajuthi et al. performed a nasal airway epithelium TWAS on asthma. The transcriptome was assessed in nasal brushing samples from 695 children. The GWAS was from a previous UK Biobank study (Ferreira et al. AJHG 2019) where the genetic associations were performed in adult- and childhood-onset asthma. Approximately 100 TWAS asthma genes were identified and more than half of them were specific to nasal samples compared to other asthma-relevant tissues in GTEx. Some known asthma genes were confirmed such as GSDMB on 17q21, IL33 on 9p24, and TSLP on 5q22. They then focused subsequent analyses and functional assays on two nasal-specific TWAS genes, namely MUC5AC and FOXA3. They concluded that cis-variants at the MUC5AC and FOXA3 asthma risk loci mediate trans-effects involving distinct mucus secretory networks.

The article is clearly written. The nasal airway transcriptomic dataset is valuable and well-suited for a TWAS on asthma. Functional assays on two TWAS genes are appreciated, but the results are based on small sample size. The TWAS results are not well-integrated with existing literature on asthma GWAS/TWAS. There are some misleading statements and the interpretation of some the results does not reflect the exploratory nature of the analyses.

Major points

1. It is misleading to use the term "airway epithelium" in the title and abstract. "Nasal airway epithelium" should be used.
2. The unique transcriptome of nasal samples compared to other GTEx tissues (Figures 1a & 1b) is likely to reflect, at least in part, technical differences in collecting samples and quantifying the transcriptome.
3. The genotype specific cell culture assays are interesting, but performed in a small sample size. Robustness of the results is a concern.
4. This statement in the abstract: "We identified an airway risk gene for 38% of reported asthma genetic risk loci,..." as well as this one in the Discussion (lines 386-388): "Moreover, our airway TWAS identified at least one significant gene for 68 of the 179 (38%) asthma risk loci identified to date by GWAS, more than for any other GTEx tissue,..." are incorrect. "179" is the number of asthma risk loci identified by Ferreira et al. (AJHG 2019) only, not all asthma GWAS to date. In addition, 179 is the sum of independent variants identified in both GWAS from Ferreira (adult- and childhood-onset asthma). There are overlapping variants and asthma risk loci in this list. What we want to know is the number of nasal TWAS genes located in distinct asthma-associated loci (and not just asthma risk loci identified by Ferreira et al.). The authors must also recognize that the same TWAS genes are found at some risk loci. For example, TSLP counts for 6 out of the 179 loci in Suppl. Table 3. Finally, a fair comparison with previous asthma TWAS should be provided.
5. It is unclear why they have focused on MUC5AC and FOXA3 vs other nasal-specific TWAS genes for in vitro assays.
6. The networks derived from trans-expression effects as well as co-expression analyses are exploratory in nature. The interpretation should be adapted accordingly.

Other comments to improve the manuscript

1. Line 84: Do you mean GSDMA? GSDMC is not on 17q21.
2. Line 118: How many are asthmatics and healthy controls?
3. The statement in lines 151-152 is not supported by Figure 1c.
4. Line 169: The abstract and Discussion indicate 108 genes, not 95.
5. Line 206: Figure 3a does not show colocalization.
6. Line 232: the GALA II cohort is introduced for the first time in the text. We understand later from the methods section that it is the same cohort used for the nasal TWAS, but this is not clear for the reader at line 232.
7. Figure 4b and Figure 5b should include sample size by genotype. We see dots, but explicit numbers would be useful.
8. Figure 5a. The COA GWAS signal does not seem statistically significant.
9. Figure 5d. It is not clear how they build a volcano plot from genome-wide trans-eQTL analysis. How fold changes were obtained? AA vs GG? Reference 66? More details are needed in text and legend.
10. Clinical characteristics of children in the GALA II study should be provided, at least in supplementary table.
11. Line 581: how many children had at least a 10 pack-year smoking history?
12. Line 587: we understand from the abstract that the TWAS was performed in n=695.
13. Line 713: provide sample size for GTEx tissues. GTEx version 8 is available with larger sample size.

Reviewer #3:

Remarks to the Author:

Dr Sajuthi and others present an interesting and well-written new manuscript entitled "Airway epithelial TWAS of childhood asthma reveals novel, genetically driven mucus pathobiology." Comparing nasal TWAS data from 695 children with GWAS data from childhood- and adult-onset asthma, they identify 108 susceptibility genes. The approach is relatively straightforward, and the manuscript is easy to read. The study populations are substantial in number and the data are well characterized.

Major comments:

The authors essentially describe the nasal epithelium as a convenience sample (no bronchoscopy needed) that is to serve as a surrogate for lower respiratory tract airways. This is a longstanding point of contention, but the authors reasonably argue about the similarities of these epithelia, and many other studies use nasal epithelial samples to represent lower airways biology. The concern here is that the nasal epithelial samples display quite different gene expression patterns than do lung samples in the presented results. Admittedly, the authors use what sounds like whole lung homogenates (i.e., containing cell types other than airway epithelial cells), but one would still anticipate detectable expression of transcripts that are expressed by both nasal and bronchial epithelial cells. The

multidimensional scaling plot and the dendrogram in Figure 1 both show vastly different expression patterns of the nasal epithelial samples and the lung samples. Then, in Figures 2B-C, there is almost no overlap in the signal from nasal epithelium and lung. Together, these observations would seem to necessitate that the authors better justify their model system as reflective of lower airways disease (even if they have found targets they believe to be relevant).

The findings that FOXA3 may drive alternate, cytokine independent metaplasia is novel and interesting. However, the paper otherwise seems to “look under the lamp post” to focus on known susceptibility genes, rather than uncovering novel targets. Perhaps the known genes are the entire story, but this kind of TWAS/GWAS work seems the ideal format to discover completely unanticipated contributors. Expansion of discussion of these unanticipated findings might improve the novelty and impact of the manuscript.

Minor comment:

In Figure 2b-c, please sequence the tissues types in the same order as they key (and, ideally, keep them the orders the same in both panels). Several of the colors are similar and determining which samples are which is more challenging than it should be.

Reviewer #1 (Remarks to the Author):

Reviewer Comment: This is an interesting study adding significantly to the understanding of the molecular mechanisms of asthma. The manuscript is well written, and the methodologies seem sound.

Author Response: Thank you for your appreciation of the findings contained in our manuscript, as well as your thoughtful review, which in addressing, we believe has improved the manuscript. See point-by-point responses below.

Major points

Reviewer Comment: My main concern relates to the interpretation/reason for the *finding that nasal gene expression is markedly different from other tissues, including lung tissue*, and performs markedly better in the TWAS comparison. There are several possible explanations for this finding that needs to be analysed and discussed in more details:

As mentioned in the introduction, the authors have previously shown that the nasal airway epithelium expresses the same genes as the bronchial airway, epithelium, with highly correlated expression levels between these sites. This seems contradictory to the finding that nasal gene expression shows no co-clustering with lung expression in GTEx.

Author Comment: We agree with the reviewer that our previous analyses have found a strong correlation between nasal and bronchial airway epithelial expression profiles.¹ Therefore, we also believe it would be strange if we found bronchial and nasal airway brushings did not cluster together in an MDS expression plot of diverse tissues. However, lung tissue is not equivalent to bronchial airway epithelium, as lung explant tissue like that collected by GTEx is mostly parenchymal tissue dominated by alveolar epithelium, capillaries, macrophages, fibroblasts, and other immune cells; with a rare embedded airway. Moreover, GTEx lung tissue collection protocols specifically direct the collection of very distal lung tissue, namely they instruct to collect lung tissue, “...1 cm below the pleural surface,” and also, “Avoid any large arteries, veins, and **bronchi**.” (https://biospecimens.cancer.gov/resources/sops/docs/GTEx_SOPs/BBRB-PR-0004-W1%20GTEx%20Tissue%20Harvesting%20Work%20Instruction.pdf). This reinforces that GTEx lung tissue would poorly reflect the bronchial airway epithelium.

Supporting the cellular complexity of lung explant tissue, single cell RNA-seq analyses have found lung tissue includes 58 different molecular cell types, only 12 of which are airway epithelial.² Our own lung explant tissue scRNA-seq analysis of 4 non-diseased donors and 9,807 cells (unpublished) found only 25.7% of total cells are airway epithelial cell types. In summary, we would not expect the top variant and expressed genes generated from the bulk RNA-seq data of lung tissue would overlap strongly with that of nasal airway epithelium, given these datasets are generated from very different cellular mixtures, and that airway epithelial cells will be a minority population within the GTEx lung tissue cell mixtures.

Regarding the TWAS analysis differences between lung and nasal, please note we found approximately as many significant lung TWAS genes (COA=82, AOA=13) as nasal (COA=88, AOA=18). Therefore, we do not mean to suggest lung tissue has no value, nor do we state this in the paper. However, the nasal airway epithelium yielded more unique TWAS genes than any other tissue, including lung tissue, a number of which are known to be important in asthma pathobiology. We believe the more homogeneous, airway-centric cellular composition of our nasal epithelial brushings vs lung tissue creates a less complex/cell-type confounded expression profile, which is more powerful for observing genetic effects on airway genes. This, coupled with that fact that the bronchial airway epithelium (which the nasal airway epithelium proxies strongly) has been

shown to play a prominent role in asthma development, is likely responsible for our strong, unique nasal TWAS gene findings.

Beyond this, we appreciate the reviewer's careful consideration of other factors potentially driving differences between our nasal results and the lung/other GTEx tissues, which we address below, point by point.

This raises the possibility that this could be due to methodological differences, eg due to platform or other differences in generating and handling the "local" and GTEx datasets. Please address this, potentially by including other eQTL sources.

Author Comment: As stated above, we believe the primary reason for the differences in clustering of the lung tissue and nasal airway epithelium are the differences in the cellular composition of the tissues. There undoubtedly will be some smaller differences between our nasal dataset and the GTEx datasets due to differences in both laboratory sequencing and sequencing analysis methodologies. However, we designed our study and analyses to minimize these potential biases. For example, both the GTEx and nasal RNA-seq datasets were generated from polyA-selected RNA-seq libraries, with sequencing performed on the Illumina platform.

Regarding the figure 1 tissue clustering analysis, we combined raw RNA-seq count data from GTEx tissues and the nasal epithelium and performed vst-normalization of this combined dataset, to minimize biases. MDS analysis was performed on the vst transformed, combined dataset using a large number of highly variable genes (n=10,000) to minimize spurious effects, which could be experienced if analyzing a small gene set. We performed our second analysis, comparing the sharing of the highest variance genes within each tissue, to avoid analysis of a combined expression dataset, altogether. Namely, we identified the highest variance genes within each dataset independently and then compared the overlap of different tissue gene sets. Therefore, this analysis should be more robust to batch differences. Since our nasal epithelium data was the only pure epithelial sample in the analysis, the unique profile of the nasal samples was expected.

Regarding the TWAS analyses: the nasal and GTEx tissues were analyzed with the same TWAS methods, so these analyses will not exhibit differences due to statistical methodology.

Nonetheless, we have added a statement to the discussion acknowledging differences in GALA vs GTEx methodology and their potential influence.

Could the better results using nasal data be due to the younger age of individuals? Please compare with GTEx participant characteristics.

Author Comment: The average age of GALA children is 14y and the 25th and 75th percentiles are 11.5y and 16.4y. For GTEx lung donors the average age is 55.8y and the 25th and 75th percentiles are 45y and 75y. Variation in the expression of genes has previously been observed between age groups.^{3,4} However, we believe the differences in TWAS results caused by these factors will be small in comparison to the dramatic effects caused by the differences in cellular composition, as described above. However, we now acknowledge the potential for age differences between GTEx and GALA influencing our results in the paper discussion.

The authors mention the larger sample size in their nasal eQTL data as a strength – how does their nasal sample size compare to the lung sample size in GTEx?

Author Comment: The GTEx lung sample size is 383 versus the 681 subjects that we used in our nasal airway eQTL/TWAS analysis. Previously the GTEx consortium published an analysis showing a linear relationship between the number of eQTLs discovered in a tissue and the sample size, at least within a range of 70-600 samples.⁵ The increased power of our larger sample size likely aided in eQTL discovery, especially for variants

driving smaller effects and for lower frequency eQTLs. However, these weaker eQTLs are less likely to yield genome-wide significant TWAS hits. Therefore, we do not believe our unique nasal TWAS findings were largely a function of sample size.

As further evidence of this, we examined a recently published TWAS analysis⁶ (Valette et al) which used lung tissue expression from 1,038 donors, a sample size which is 52% larger than our nasal analysis sample set. The Valette TWAS analysis was also performed using genetic data from the UKBiobank, but the Valette GWAS was performed with 56,167 asthma cases and 352,255 controls, a larger sample set than used in our study. To be clear, the Valette GWAS/TWAS did not stratify by childhood and adult onset asthma. The Valette asthma lung TWAS analysis only identified 55 significant genes, versus the 82 TWAS genes identified in the GTEx Lung COA TWAS and the 13 genes identified in the GTEx Lung AOA TWAS, despite the smaller GTEx expression datasets. Therefore, we surmise that the GTEx Lung TWAS analysis we conducted was not appreciably restricted in the number of genes identified because of power.

In terms of the effect of the larger Valette sample size on the identification of genes we found to be nasal-specific in our TWAS: only 1 of the 11 AOA TWAS genes and 3 of the 51 COA TWAS genes we found to be nasal-specific in comparison to GTEx tissues, were identified as significant asthma TWAS genes in the larger Valette lung dataset. Moreover, none of the important nasal-specific TWAS genes discussed in paper, *FOXA3*, *IL33*, *MUC5AC*, *MUC2*, were identified in the Valette TWAS.

Therefore, we do not think the sample size differences are responsible for the majority of the unique genes identified in the nasal vs, lung TWAS analyses, but rather the limited airway epithelial component of the lung expression datasets, as discussed above. However, we now acknowledge sample size differences between the GTEx lung and GALA nasal analyses in the discussion and reference the larger lung TWAS publication results in the discussion section as well.

Another important point is that a large proportion of individuals providing nasal eQTL data were asthmatics. This is mentioned shortly in the discussion but not analytically. Probably very few individuals in the GTEx dataset were asthmatics? First, I wonder if this could lead to some inflation of the eQTL signals as a result of differential expression between asthmatics and controls? Second, the possibility that the improved performance of the nasal data is due to more relevant eQTL data in asthmatics is important for directing future studies. The effect of having asthmatics in the nasal eQTL data should therefore be analyzed, eg by stratified analyses, at least in a targeted way focusing on the most prominent TWAS findings.

Author Comment: We do not have data on the number of asthmatics in the GTEx cohort, but it is a safe assumption that the percentage of asthmatics in our GALA nasal dataset (63.7%) is considerably more than the percentage of asthmatics contained in the lung or any other GTEx tissue dataset. As to whether analyzing a mix of asthmatics and control subjects could have resulted in increased signals, our analysis was performed on common variants (>5%) and well-expressed genes, both of which were well-represented among both asthmatics and control groups. While the expression level of some genes and the allele frequency of some variants did vary between asthmatic and healthy groups, the genetic models built for the TWAS analysis are accounted for asthma status, to correct for any effects driven by disease status. Beyond this, large GWAS studies for asthma have made clear that most genetic risk conferred by common variants (>5% allele frequency) exert small effects on disease risk (odds ratios 1.01-1.5); therefore, we don't expect that the small allele frequency differences between asthmatic and control groups would substantially alter our ability to identify eQTLs important for asthma risk.

One scenario which could result in additional eQTL discoveries in a cohort enriched for asthma would be if the relationship between gene expression and genotype differed by asthma status (ex. if there is an eQTL present in the asthma group, but not in controls). As suggested, we explore this issue by examining the most

prominently featured TWAS gene associations reported in the paper (*MUC5AC*, *FOXA3*, *IL33*, *IKZF3*, *TSLP*, and *GSDMB*) with respect to asthma status. Specifically, we plot expression of these genes by the strongest eQTL variant for the gene, which drives the TWAS association, among asthmatics and controls separately. As can be seen below, the identified eQTL variant exerts a strong, significant effect on expression, in the same direction, among both asthmatic (n=434) and control groups (n=247), despite the smaller size of the control group. In fact, the effect size of the eQTL variant is nearly identical between groups for all six TWAS genes. Also note we already discuss the potential influence of having asthmatics in our analysis set in the paper discussion.

If the stronger results using nasal data is a true biological finding, does it suggest that the nasal epithelium is more important than the lung in asthma inception?

Author Comment: We believe the excellent and unique performance of the nasal airway brushings are both a function of importance in disease and cellular composition. As detailed above, a definite strength of nasal epithelium in these analyses is in its ability to proxy expression of the bronchial airway epithelium, dysfunction of which has been shown to play an important role in asthma development. Moreover, asthma is a gene-environment disease, and this interaction first occurs at the nasal airway epithelium. For example, inhaled allergens first stimulate immune/epithelial cells in the nose, respiratory viruses first and primarily infect the nasal epithelium. These upper airway stimuli can trigger both systemic and lower airway effects, therefore function of nasal airway epithelium is likely important in its own right for asthma development. Additionally, RNA-seq of nasal brushings provide a relatively pure epithelial sample that is less cellularly complex and confounded by multiple independent cell type expression profiles, relative to lung tissue, rendering it easier to identify gene-eQTL effects, especially that differ by cell type. Lung tissue expression profiles will be dominated by both alveolar and interstitial cell expression profiles, neither of which are suspected to be prominently

involved in asthma development. That being said we still identified 82 COA and 13 AOA lung TWAS genes, therefore lung tissue definitely has value in deciphering asthma risk genes.

In summary, with the added revisions, we believe our analysis and interpretation of the differences in nasal and lung tissue results is well-justified and balanced. Thank you for your insightful comments in this regard.

Regarding the MUC5AC trans-eQTL analysis:

The authors suggest that the MUC5AC eQTL increases asthma risk by not only increasing MUC5AC expression, but also through activating networks of genes needed for the production and secretion of MUC5AC+ mucus from airway secretory cells. However, given the biological property of MUC5AC, as a mucin and not a transcription factor or immune mediator, it seems equally plausible that the trans-effects observed could be responses to altered mucus properties, without any effect on asthma risk. Without any direct evidence that these trans-effects are related to asthma risk, I think a more careful interpretation of these trans-signals is warranted.

Author Comment: We agree with the reviewer that these trans-expression effects are not due to MUC5AC acting as a transcription factor, nor do we suggest this in the manuscript. We likewise agree that a plausible explanation for these trans-effects is the altered mucus “feeding-back” on the epithelium, triggering changes in transcription that allow altered processing and secretion of the higher MUC5AC-content mucus. It is also possible the increased intracellular MUC5AC levels alter cellular signaling and transcription. As to whether the potential causation of the MUC5AC risk locus is through increased MUC5AC alone or increased MUC5AC in combination with associated trans-effect is unclear. We raise this as a possibility, which it definitely is, in a conclusion statement to the MUC5AC results section. However, since this is speculation more suitable for the discussion, we have removed the reference to the trans-effect possibly “increasing asthma risk” from the results. We have also revised statements in the discussion to simply state the MUC5AC asthma risk eQTL is associated with trans-effects, and to make clear we don’t know if these trans-effects are causal in the eQTL-asthma association.

Minor point

The authors identified 8 COA TWAS genes and a single AOA TWAS gene that were not within a Mb of any GWAS risk variant. While most of the other TWAS genes have previously been suspected as causal genes underlying asthma loci, these 9 are the most novel potential risk genes and should be discussed.

Author Comment: Please note one of these genes is *FOXA3*, which we study and discuss extensively. However, we have added the gene names to the results and commentary covering a two other of the most interesting genes to the discussion.

Reviewer #2 (Remarks to the Author):

Sajuthi et al. performed a nasal airway epithelium TWAS on asthma. The transcriptome was assessed in nasal brushing samples from 695 children. The GWAS was from a previous UK Biobank study (Ferreira et al. AJHG 2019) where the genetic associations were performed in adult- and childhood-onset asthma. Approximately 100 TWAS asthma genes were identified and more than half of them were specific to nasal samples compared to other asthma-relevant tissues in GTEx. Some known asthma genes were confirmed such as *GSDMB* on 17q21, *IL33* on 9p24, and *TSLP* on 5q22. They then focused subsequent analyses and functional assays on two nasal-specific TWAS genes, namely *MUC5AC* and *FOXA3*. They concluded that cis-variants at the *MUC5AC* and *FOXA3* asthma risk loci mediate trans-effects involving distinct mucus secretory networks.

The article is clearly written. The nasal airway transcriptomic dataset is valuable and well-suited for a TWAS on asthma. Functional assays on two TWAS genes are appreciated, but the results are based on small sample size. The TWAS results are not well-integrated with existing literature on asthma GWAS/TWAS. There are some misleading statements and the interpretation of some the results does not reflect the exploratory nature of the analyses.

Author Comment: We thank the reviewer for the thorough assessment of the manuscript, which in addressing has improved the manuscript. We address all points below.

Major points

1. It is misleading to use the term “airway epithelium” in the title and abstract. “Nasal airway epithelium” should be used.

Author Comment: We have modified the title and abstract as suggested.

2. The unique transcriptome of nasal samples compared to other GTEx tissues (Figures 1a & 1b) is likely to reflect, at least in part, technical differences in collecting samples and quantifying the transcriptome.

Author Comment: We do not believe technical differences are responsible for most of these effects, as explained in detail, in responses to reviewer 1. However, we have added this a potential limitation in the discussion.

3. The genotype specific cell culture assays are interesting, but performed in a small sample size. Robustness of the results is a concern.

Author Comment: We believe these experiments add greatly to the manuscript and in vitro human validation/exploration of genetic effects is rare in the literature, adding to their value. Moreover, all reported cell culture results reached statistical significance, and there is great conceptual coherence between the results for multiple independent assays (in vivo and in vitro expression effects align, in vitro protein effects align with expression effects, direction of gene/protein effects conceptually align with asthma risk, trans-effects mechanistically align with an increase in mucin production), adding to our confidence in them. However, we agree these results will need to be replicated and explored further in other sample sets to achieve higher confidence and understanding. Therefore, we have added a statement of limitation to the discussion, to emphasize the need for further replication and study.

4. This statement in the abstract: “We identified an airway risk gene for 38% of reported asthma genetic risk loci,...” as well as this one in the Discussion (lines 386-388): “Moreover, our airway TWAS identified at least one significant gene for 68 of the 179 (38%) asthma risk loci identified to date by GWAS, more than for any other GTEx tissue,...” are incorrect. “179” is the number of asthma risk loci identified by Ferreira et al. (AJHG 2019) only, not all asthma GWAS to date. In addition, 179 is the sum of independent variants identified in both GWAS from Ferreira (adult- and childhood-onset asthma). There are overlapping variants and asthma risk loci in this list.

We agree it was confusing to combine the COA and AOA risk variants into one pool. We have revised our text in the abstract and the discussion (see below) to separate the percentages of nasal TWAS genes observed for the independent COA and AOA risk loci. We have also now clarified that our percentages are with reference to the asthma loci identified by the Ferreira et al UKBiobank GWAS. Regarding our use of independent variants, rather than loci: firstly, we used independent variants, since that is what was reported by the Ferreira et al paper, secondly, how one consolidates independent risk variants, nearby on the same chromosome, into loci

is not straightforward. However, we take the reviewers point that consolidation of nearby variants into loci avoids counting TWAS genes multiple times as underlying independent risk variants in close proximity to one another. Therefore, we condense Ferreira et al independent variants into loci that are within 1Mb of one another. In doing so, Ferreira et al UKBiobank GWAS identifies 89 COA loci and 40 AOA loci.

Our revised abstract text now reads:

“We identified an airway risk gene for 36% of COA and 33% of AOA GWAS loci identified in the same UKBiobank dataset, including airway-specific effects for the strongest asthma risk loci (IKZF3 at 17q21 and IL33).”

Our revised discussion text now reads:

“Moreover, our airway TWAS identified at least one significant gene for 32 of the 89 COA risk loci (36%) and 13 of the 40 AOA risk loci (33%) identified by the Ferreira et al GWAS analysis of UKBiobank data used in our TWAS analysis, affirming the importance of airway epithelial dysfunction in asthma development.”

What we want to know is the number of nasal TWAS genes located in distinct asthma-associated loci (and not just asthma risk loci identified by Ferreira et al.).

We believe it is beyond the scope of this article to reference our TWAS genes to every asthma-associated loci published to date. It would be difficult to decide which studies to include, if they were properly powered, and what are distinct loci across different study populations. Rather, we think it is highly appropriate and interpretable, when our question is how much of asthma genetic risk is even potentially conferred by changes in nasal gene expression, to assess GWAS loci generated in the same dataset used for the TWAS analyses. All data used is provided in the manuscript, which will allow readers to analyze any GWAS dataset desired.

The authors must also recognize that the same TWAS genes are found at some risk loci. For example, TSLP counts for 6 out of the 179 loci in Suppl. Table 3.

As noted above, to address this, we first separate the 179 independent risk variants into 123 COA and 56 AOA independent risk variants. We then condense the independent risk variants into 89 COA and 40 AOA risk loci. We then calculate the percentage of COA loci and AOA loci with a nasal TWAS gene in proximity to the loci, which is 36% and 33%, respectively. With these revisions, no genes are double counted.

Finally, a fair comparison with previous asthma TWAS should be provided.

The goal of our study was to perform a TWAS analysis of nasal airway epithelium brushing data and to compare these results to TWAS results generated from other tissues. Using publicly available GTEx consortium data we have tried to do this in the most rigorous and comparable fashion, generating TWAS data by the same method and with the same GWAS data used for our nasal TWAS. At time of submission the Pividori et al TWAS⁷ was the only significant asthma TWAS analysis we were aware of. This paper was cited. The Pividori et al and Ferreira et al studies both used highly similar COA/AOA GWAS analyses of the same UKBiobank population. Moreover, the Pividori et al TWAS also used expression data from GTEx. Therefore, we believe it would be largely redundant to compare our results to this paper, when we already compare our UKBiobank/GTex based TWAS analysis to our nasal TWAS analysis. However, now we add another citation to this paper in the results section, when we introduce our GTEx TWAS and note its similarity to our analysis. Since our paper entered peer-review, another larger GTEx lung tissue based TWAS analysis has been performed (as noted above), we now reference these results in the discussion.

5. It is unclear why they have focused on MUC5AC and FOXA3 vs other nasal-specific TWAS genes for in vitro assays.

Author Comment: Mucus obstruction is one of the most prominent, yet poorly understood aspects of asthma pathobiology, most especially with regard to genetic drivers of this pathobiology. Moreover, the effect of these genes is clearly through modification of the airway epithelium, which is the central tissue in our analyses. Therefore, both due to need within the field and positioning on our study, we decided to pursue detailed exploration of these genes.

6. The networks derived from trans-expression effects as well as co-expression analyses are exploratory in nature. The interpretation should be adapted accordingly.

Author Comment: We agree the MUC5AC trans-expression network is exploratory and we have added the term “exploratory” to the introduction of this network. The co-expression networks were determined agnostically, based on a highly powered co-expression analysis in a large dataset, and are described by functional enrichments and known biology. We ran statistically valid enrichment analyses of trans-genes. We do not agree that these are exploratory analyses. However, as a gene group we now refrain from calling the trans-genes alone a network, but rather that they are enriched among networks. Moreover, we do state the overall hypotheses generated by our analyses will need to be further explored in the discussion section.

Other comments to improve the manuscript

1. Line 84: Do you mean GSDMA? GSDMC is not on 17q21.

Author Comment: Thank you for catching this error. We have corrected in the text accordingly.

2. Line 118: How many are asthmatics and healthy controls?

Author Comment: We have added the number of asthmatics (N=434) and healthy controls (N=247) with both nasal RNA-seq and WGS sequencing data to the text.

3. The statement in lines 151-152 is not supported by Figure 1c.

Author Comment: We have removed “figure 1c” from the text.

4. Line 169: The abstract and Discussion indicate 108 genes, not 95.

Author Comment: Thank you for catching this error. We have modified the text accordingly.

5. Line 206: Figure 3a does not show colocalization.

Author Comment: Thank you for catching this error. We have removed reference to “figure 3a” from the text.

6. Line 232: the GALA II cohort is introduced for the first time in the text. We understand later from the methods section that it is the same cohort used for the nasal TWAS, but this is not clear for the reader at line 232.

Author Comment: We have modified the text to introduce the use of the GALA II cohort in the introduction section.

7. Figure 4b and Figure 5b should include sample size by genotype. We see dots, but explicit numbers would be useful.

Author Comment: We have added the sample size for figure 4b and figure 5b in their respective figure legends.

4b. N; T2 low AA: 67; T2 low GA: 166; T2 low GG: 91; T2 high AA: 77; T2 high GA: 180; T2 high GG: 100.

5b. N; T2 low AA: 18; T2 low GA: 116; T2 low GG: 190; T2 high AA: 29; T2 high GA: 135; T2 high GG: 193

8. Figure 5a. The COA GWAS signal does not seem statistically significant.

Author Comment: The reviewer is correct. FOXA3 is a gene that we found to be TWAS significant even though the FOXA3 locus itself is not GWAS significant. In figure 5a, we are trying to show the colocalization between the COA GWAS signals and nasal epithelium eQTL signals around the FOXA3 locus.

9. Figure 5d. It is not clear how they build a volcano plot from genome-wide trans-eQTL analysis. How fold changes were obtained? AA vs GG? Reference 66? More details are needed in text and legend.

Author Comment: To clarify how we obtain the fold change, we have added this sentence to figure 5d: "LFC estimate of each gene is obtained from limma⁸ by regressing gene expression values on additively coded rs8103278 genotype (A/A: 0; G/A: 1; G/G: 2)".

10. Clinical characteristics of children in the GALA II study should be provided, at least in supplementary table.

Author Comment: We have added this information into Supplementary Table 13

11. Line 581: how many children had at least a 10 pack-year smoking history?

Author Comment: "at least 10 pack-year of smoking history" is one of GALA II exclusionary criteria, hence none of the children in GALA II have a history of smoking of more than 10 pack-year.

12. Line 587: we understand from the abstract that the TWAS was performed in n=695.

Author Comment: There were 695 donors with RNA-seq data but only 681 donors have paired genotype data. We have changed the number to 681 to clarify this.

13. Line 713: provide sample size for GTEx tissues. GTEx version 8 is available with larger sample size.

Author Comment: We have added these numbers to the manuscript. At the time of analysis only GTEx v7 was available.

1. Poole A, Urbanek C, Eng C, et al. Dissecting childhood asthma with nasal transcriptomics distinguishes subphenotypes of disease. *J Allergy Clin Immunol* 2014;133:670-8 e12.
2. Travaglini KJ, Nabhan AN, Penland L, et al. A molecular cell atlas of the human lung from single-cell RNA sequencing. *Nature* 2020;587:619-25.
3. Lu T, Pan Y, Kao SY, et al. Gene regulation and DNA damage in the ageing human brain. *Nature* 2004;429:883-91.
4. Mele M, Ferreira PG, Reverter F, et al. Human genomics. The human transcriptome across tissues and individuals. *Science* 2015;348:660-5.
5. Consortium GT. The GTEx Consortium atlas of genetic regulatory effects across human tissues. *Science* 2020;369:1318-30.
6. Valette K, Li Z, Bon-Baret V, et al. Prioritization of candidate causal genes for asthma in susceptibility loci derived from UK Biobank. *Commun Biol* 2021;4:700.
7. Pividori M, Schoettler N, Nicolae DL, Ober C, Im HK. Shared and distinct genetic risk factors for childhood-onset and adult-onset asthma: genome-wide and transcriptome-wide studies. *Lancet Respir Med* 2019;7:509-22.
8. Ritchie ME, Phipson B, Wu D, et al. limma powers differential expression analyses for RNA-sequencing and microarray studies. *Nucleic Acids Res* 2015;43:e47.

Reviewers' Comments:

Reviewer #1:

Remarks to the Author:

The authors have done a good job addressing the reviewers' concerns.

The authors' main explanation for the different clustering and better TWAS performance in nasal compared to lung tissue is that lung tissue is more heterogenous with a small proportion of epithelial cells and therefore a poorer reflection of bronchial airway epithelium. I suggest expanding on this explanation in the discussion (eg around line 451) including the details on sampling of lung tissue in GTEx avoiding bronchi and the references to the studies demonstrating heterogeneity in cell populations in lung tissue. All reviewers suspected some of the nasal specific results to be due to methodological issues and many readers are likely to do the same and will therefore benefit from such expanded discussion.

Responses to reviewer 3 comments are lacking.

Reviewer #2:

Remarks to the Author:

Line 466: "a recently published GTEx-lung/UKBiobank-asthma TWAS" is incorrect. The main lung eQTL dataset (n=1038) used by Valette et al. was not from GTEx.

Reviewer #3:

Remarks to the Author:

The revised manuscript from Sajuthi and colleagues is only modestly changed in content, but is improved in its handling of the nasal vs. lung sample concern raised by both reviewers. Based on the authors' comments, asthmatic data are not available for the GTEx cohort. So, direct assessment of the impacts of asthma prevalence on eQTL data cannot be performed. The revised inferences regarding MUC5AC are more appropriately stated.

In all, the manuscript is improved and there are clear limits to what can be additionally done with the available datasets. The presented data have the capacity to advance the field, and the methodology is as sound as can be achieved with the available datasets.

Reviewer #1 (Remarks to the Author):

Reviewer Comment: The authors have done a good job addressing the reviewers' concerns.

Response: Thank you.

Reviewer Comment: The authors' main explanation for the different clustering and better TWAS performance in nasal compared to lung tissue is that lung tissue is more heterogenous with a small proportion of epithelial cells and therefore a poorer reflection of bronchial airway epithelium. I suggest expanding on this explanation in the discussion (eg around line 451) including the details on sampling of lung tissue in GTEx avoiding bronchi and the references to the studies demonstrating heterogeneity in cell populations in lung tissue. All reviewers suspected some of the nasal specific results to be due to methodological issues and many readers are likely to do the same and will therefore benefit from such expanded discussion.

Response: We agree, and have now added this explanation and references to the discussion.

Reviewer Comment: Responses to reviewer 3 comments are lacking.

Response: We answered all comments provided to us in the prior review, point by point. Reviewer #3 below indicated the adequacy of our responses.

Reviewer #2 (Remarks to the Author):

Reviewer Comment: Line 466: "a recently published GTEx-lung/UKBiobank-asthma TWAS" is incorrect. The main lung eQTL dataset (n=1038) used by Valette et al. was not from GTEx.

Response: Thank you for catching this error, we have removed the GTEx reference and now cite the original lung eQTL paper here as well.

Reviewer #3 (Remarks to the Author):

Reviewer Comment: The revised manuscript from Sajuthi and colleagues is only

modestly changed in content, but is improved in its handling of the nasal vs. lung sample concern raised by both reviewers. Based on the authors' comments, asthmatic data are not available for the GTEx cohort. So, direct assessment of the impacts of asthma prevalence on eQTL data cannot be performed. The revised inferences regarding MUC5AC are more appropriately stated.

Response: We agree the revised manuscript now more adequately deals with the analytical complexities raised by the reviewers. Thank you again for your comments.

Reviewer Comment: In all, the manuscript is improved and there are clear limits to what can be additionally done with the available datasets. The presented data have the capacity to advance the field, and the methodology is as sound as can be achieved with the available datasets.

Response: Thank you for recognizing the value of our work.